# Loss of NPC1 enhances phagocytic uptake and impairs lipid trafficking in microglia

Alessio Colombo[1,10], Lina Dinkel[1], Stephan A. Müller [1,2], Laura Sebastian Monasor[1], Martina Schifferer [3], Ludovico Cantuti-Castelvetri [1,4], Jasmin König[1,5], Lea Vidatic[6], Tatiana Bremova-Ertl[7,8], Andrew P. Lieberman[9], Silva Hecimovic[6], Mikael Simons[1,3,4], Stefan F. Lichtenthaler[1,2,3], Michael Strupp[7], Susanne A. Schneider[7] & Sabina Tahirovic [1✉]

Niemann-Pick type C disease is a rare neurodegenerative disorder mainly caused by mutations in *NPC1*, resulting in abnormal late endosomal/lysosomal lipid storage. Although microgliosis is a prominent pathological feature, direct consequences of NPC1 loss on microglial function remain not fully characterized. We discovered pathological proteomic signatures and phenotypes in NPC1-deficient murine models and demonstrate a cell autonomous function of NPC1 in microglia. Loss of NPC1 triggers enhanced phagocytic uptake and impaired myelin turnover in microglia that precede neuronal death. *Npc1−/−* microglia feature a striking accumulation of multivesicular bodies and impaired trafficking of lipids to lysosomes while lysosomal degradation function remains preserved. Molecular and functional defects were also detected in blood-derived macrophages of NPC patients that provide a potential tool for monitoring disease. Our study underscores an essential cell autonomous role for NPC1 in immune cells and implies microglial therapeutic potential.

[1] German Center for Neurodegenerative Diseases (DZNE) Munich, Munich, Germany. [2] Neuroproteomics, School of Medicine, Klinikum rechts der Isar, Technical University of Munich, Munich, Germany. [3] Munich Cluster for Systems Neurology (SyNergy), Munich, Germany. [4] Institute of Neuronal Cell Biology (TUM-NZB), Technical University of Munich, Munich, Germany. [5] Faculty of Chemistry, Technical University Munich, Garching, Germany. [6] Division of Molecular Medicine, Ruder Boskovic Institute, Zagreb, Croatia. [7] Department of Neurology, Ludwig-Maximilians University, Munich, Germany. [8] Department of Neurology, University Hospital Bern, Bern, Switzerland. [9] Department of Pathology, University of Michigan, Ann Arbor, MI, USA. [10] These authors contributed equally: Alessio Colombo, Lina Dinkel. ✉email: Sabina.Tahirovic@dzne.de

Niemann-Pick type C (NPC) disease is a rare lipid storage disorder (LSD) with heterogeneous presentations, including neurological, systemic and psychiatric symptoms. NPC patients suffer from ataxia, supranuclear saccade and gaze palsy and epileptic seizures due to progressive neurodegeneration, leading to premature death[1–3]. Psychiatric symptoms comprise bipolar disorder, schizophrenia-like psychosis or major depression[4]. NPC mainly manifests during childhood, although juvenile and adult cases have also been reported[1,3–6]. Approximately 95% of the patients carry autosomal recessive mutations in the *NPC1* gene, with the remaining cases being caused by mutations in the *NPC2*[3]. The *NPC1* gene encodes for a transmembrane protein and the *NPC2* for a soluble protein that are jointly responsible for the egress and recycling of lipoprotein-derived cholesterol from late endosomes/lysosomes toward other cell compartments like the endoplasmic reticulum (ER) or plasma membrane[7–9]. Impairment of this lipid trafficking route causes an abnormal accumulation of unesterified cholesterol and other lipids (e.g., glycosphingolipids, sphingomyelin, and sphingosine) in late endosomal/lysosomal compartments, resulting in endosomal/lysosomal dysfunction[10,11].

A mouse model that carries a spontaneous loss of function mutation within the *Npc1* gene (deletion of 11 out of its 13 transmembrane domains, BALB/cNctr-Npc1$^{m1N}$/J, further abbreviated as *Npc1*$^{-/-}$)[12,13] reliably features early-onset human pathology, including neurodegeneration of vulnerable NPC regions, such as Purkinje cells in the cerebellum or other neurons in the thalamus[14,15]. The cortex and hippocampus appear less affected in NPC[16]. Behavioral defects, such as mild cerebellar ataxia and tremor, can be detected in *Npc1*$^{-/-}$ mice at 6 weeks of age and become more prominent by 8 weeks. Severe ataxia, difficulties in food and water uptake and weight loss appear by 10-12 weeks of age (humane endpoint)[17].

The molecular mechanism responsible for neuronal death in NPC is still not fully understood. It has been proposed that accumulation of lipids, particularly sphingosine, can induce an imbalance in calcium homeostasis and affect lysosomal trafficking[18]. Additionally, lipid accumulation within lysosomes and mitochondrial membranes may induce oxidative stress[19,20]. Different studies linked NPC1 dysfunction to alterations in mammalian target of rapamycin complex 1 (mTORC1) and microtubule-associated proteins 1A/1B light chain 3B (LC3) signaling, suggesting that autophagy might be compromised in NPC[21–25]. Although peripheral organs such as liver and spleen are affected by the disease, restoring NPC1 function in the central nervous system (CNS) only prevents neurodegeneration and premature lethality of the *Npc1*$^{-/-}$ mouse[26]. However, restoring *Npc1* expression in neurons only does not fully rescue the phenotype and still results in lethality, suggesting that NPC1 is functionally important in other brain cells as well[27–30]. Noteworthy, the *Npc1* gene is ubiquitously expressed throughout the brain[31], with particularly high expression in oligodendrocytes and microglia[32]. Accordingly, it was shown that NPC1 function is crucial for correct maturation of oligodendrocyte progenitor cells (OPCs) and the maintenance of the existing myelin[33,34].

Microglia, as the resident immune cells of the CNS, regulate brain homeostasis by orchestrating essential physiological processes like myelination and synaptogenesis[35], but also actively contribute to pathophysiology of neurodegenerative disorders[36–41]. Gene expression studies including Alzheimer's disease (AD), amyotrophic lateral sclerosis, fronto-temporal lobar degeneration or multiple sclerosis have underscored microglial diversity and delineated homeostatic and disease signatures, often assigned as "microglial neurodegenerative phenotype" or "disease-associated microglia" (DAM)[36–38,42–46]. Loss of *Npc1* is also associated with a massive microgliosis[47–49], and alterations of transcriptomic signatures were reported in microglia isolated from symptomatic

*Npc1*$^{-/-}$ mice[50]. However, as microglial pathology can be triggered by neurodegenerative environment and consequences of a specific loss of NPC1 in microglia have not been reported, it is still debated whether microglial activation plays a causative role in NPC pathology and merits therapeutic investigation[11,50–53]. Although beneficial effects of reducing microglial activation were reported[17,50,54], cell culture experiments suggested that *Npc1*$^{-/-}$ microglia do not directly trigger neurotoxicity[55] and microglial ablation in an NPC murine model was not beneficial[52]. Nevertheless, significant changes in inflammatory markers were reported in both pre-symptomatic murine model and NPC patients, suggesting that immune response could be a precocious phenomenon preceding neuronal loss[47,48,56–58]. Taking together the fundamental contribution of microglia during brain development and pronounced NPC pathology at childhood age, early microglial dysregulation could have profound pathological consequences. Thus, deciphering microglial contribution to NPC neuropathology is of key importance to judge whether modulation of the inflammatory response bears therapeutic potential.

In the present study, we demonstrated that loss of NPC1 can lock microglia in a disease state, largely compromising their physiological functions. Additionally, we characterized a cholesterol-triggered lipid trafficking defect in *Npc1*$^{-/-}$ microglia and discovered that accumulation of undigested lipid material is rather a consequence of impaired trafficking to lysosomes, as lysosomal proteolytic function remains preserved. Furthermore, we demonstrated that macrophages of NPC patients recapitulate major molecular and functional defects observed in the brain microglia, and thus present a valuable model for biomarker studies and clinical monitoring. Our study underscores an essential cell autonomous role for NPC1 in microglia and implies their therapeutic potential.

## Results

**Loss of NPC1 induces cholesterol storage in microglia and triggers disease associated proteomic and functional signatures.** Symptomatic *Npc1*$^{-/-}$ mice (8 weeks) display a severe degeneration in the cerebellum, particularly affecting the Purkinje cell layer which was visualized by Calbindin immunostaining (Supplementary Fig. 1a), as already reported[14,15]. Increased immunoreactivity of a myeloid-specific lysosomal marker CD68 (Supplementary Fig. 1a) revealed microglial cells with amoeboid morphology[50], suggesting their active phagocytic status[59–61]. However, regions like cortex (Supplementary Fig. 1b) and hippocampus (Supplementary Fig. 1c) also showed pronounced CD68 immunoreactivity in symptomatic *Npc1*$^{-/-}$ mice, but without detectable alterations of neuronal marker NeuN, thus excluding overt neuronal loss.

It is known that *Npc1* is highly expressed in microglia[32], implicating a cell autonomous function in immune cells. To test this hypothesis, we isolated primary cerebral microglia from symptomatic *Npc1*$^{-/-}$ mice and cultured them in vitro. Using the cholesterol binding dye Filipin[62], we found increased intracellular cholesterol content in *Npc1*$^{-/-}$ microglia compared to WT (Fig. 1a). Co-staining with CD68 showed that most of the cholesterol load in *Npc1*$^{-/-}$ microglia was localized within late endosomes/lysosomes. Thus, cultured *Npc1*$^{-/-}$ microglia isolated from the neurodegenerative brain environment still display a cholesterol storage phenotype as reported[55]. This result suggests that, in addition to neurons, NPC1 likely exerts a cell autonomous function also in microglia and microglial phenotype should not be considered only as a bystander of neurodegeneration[28,53].

Findings in AD models revealed that under pathological conditions microglia can switch from a resting homeostatic to a DAM state that is characterized by distinct transcriptomic

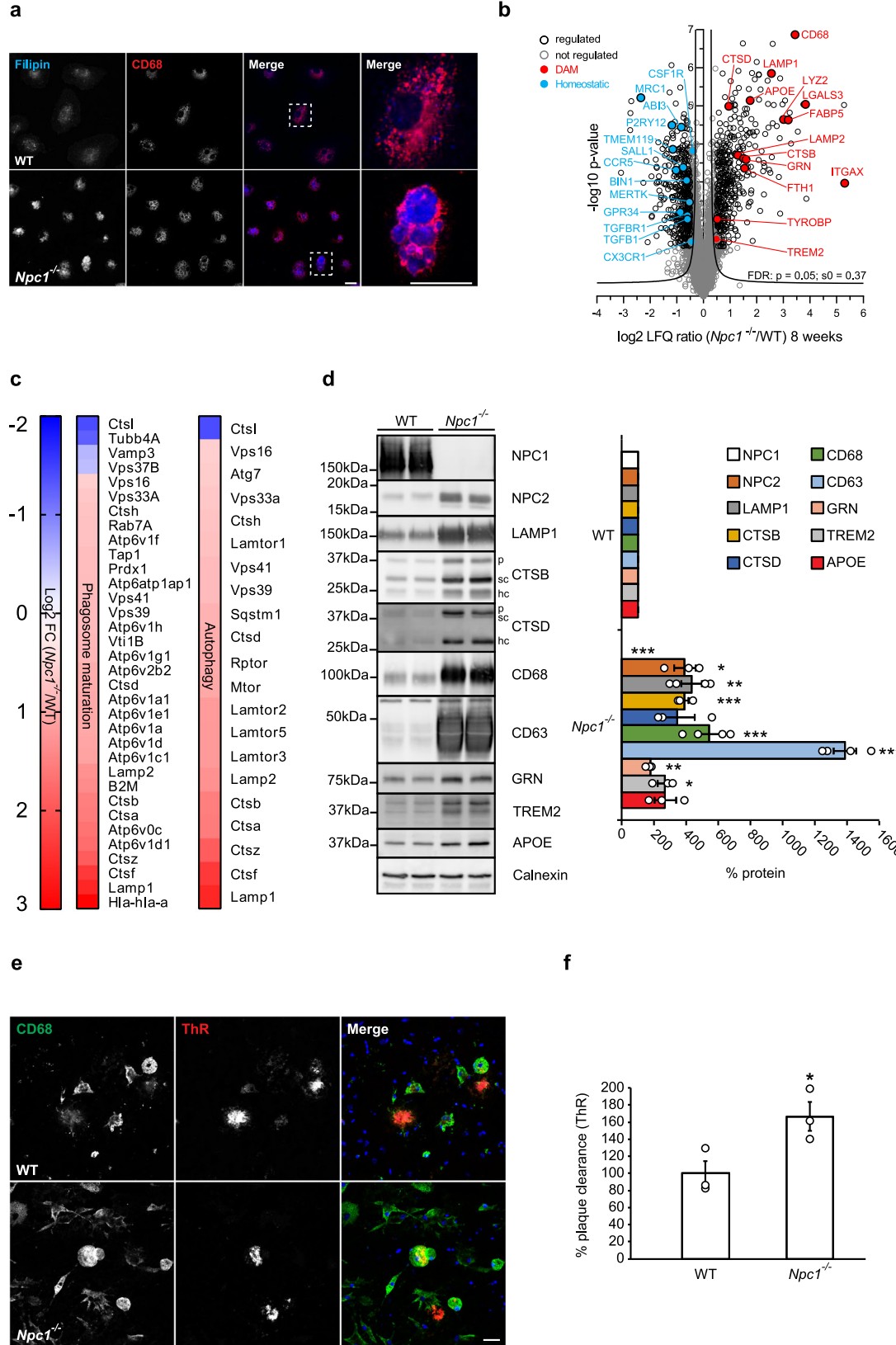

signatures[37,63]. We profiled the proteome of cerebral microglia acutely isolated from 8 weeks old *Npc1*[−/−] and WT mice using mass spectrometry (MS). Quality control of microglia enriched fraction was performed via western blot analysis using specific markers for microglia (Iba1), neurons (Tuj1), astrocytes (GFAP) and oligodendrocytes (CNPase) (Supplementary Fig. 1d). Proteomic profiles of microglia isolated from 8 weeks old mice revealed major differences between *Npc1*[−/−] and WT, featuring disease associated signatures (Fig. 1b, Supplementary Data 1)[37,38]. Among significantly upregulated proteins, we identified TREM2,

**Fig. 1 Loss of NPC1 induces microglial molecular and functional changes in symptomatic *Npc1*<sup>−/−</sup> mice. a** Immunocytochemistry of cultured primary microglia. Microglia were analyzed from three independent experiments ($n = 3$). Cholesterol was visualized using Filipin (blue). *Npc1*<sup>−/−</sup> cells show increased Filipin levels compared to WT, demonstrating cholesterol accumulation. Boxed regions are enlarged in right panels and show cholesterol accumulation within CD68 positive compartments (red) of *Npc1*<sup>−/−</sup> microglia. Scale bars: 25 μm. **b** Proteome analysis of acutely isolated microglia from 8-weeks old WT and *Npc1*<sup>−/−</sup> littermates reveals significantly decreased homeostatic and increased DAM markers in *Npc1*<sup>−/−</sup> microglia. The negative log10 transformed *p*-value of each protein is plotted against its average log2 transformed LFQ ratio between *Npc1*<sup>−/−</sup> and WT microglia. The hyperbolic curves indicate the threshold for a permutation-based FDR correction for multiple hypotheses. Significantly changed proteins (*p*-value less than 0.05 and FDR corrected) with a log2 fold change larger than 0.5, or smaller than −0.5 are encircled in black (regulated), not regulated proteins are encircled in gray and selected homeostatic (downregulated) and DAM (upregulated) proteins are highlighted in blue and in red, respectively. Microglia were analyzed from three independent experiments ($n = 3$). **c** Phagosome maturation and autophagy are the most affected pathways upon NPC1 loss of function. The heatmap shows the average log2 transformed LFQ ratio between WT and *Npc1*<sup>−/−</sup> microglia from significantly regulated proteins involved in phagosome maturation and autophagy according to IPA. **d** Validation of MS data via western blot analysis and corresponding quantification. Representative immunoblots of acutely isolated WT and *Npc1*<sup>−/−</sup> microglia show increased levels of DAM proteins (NPC2, LAMP1, CTSB, CTSD, CD68, CD63, GRN, TREM2, and APOE). Calnexin was used as loading control. For cathepsins: p = pro-form; sc = single chain form; hc = heavy chain form. Proteins were quantified by densitometry (ImageJ—NIH). At least three independent experiments were performed. Values were normalized on WT control and represent mean ± SEM (unpaired two-tailed Student's *t*-test). NPC1 ($n = 4$, $p = 5.5 \times 10^{-8}$); NPC2 ($n = 3$, $p = 0.011$); LAMP1 ($n = 4$, $p = 0.0019$); CTSB ($n = 3$, $p = 5 \times 10^{-4}$); CTSD ($n = 3$, $p = 0.08$); CD68 ($n = 4$, $p = 6 \times 10^{-4}$); CD63 ($n = 4$, $p = 1.5 \times 10^{-6}$); GRN ($n = 3$, $p = 0.003$); TREM2 ($n = 3$, $p = 0.015$); APOE ($n = 3$, $p = 0.068$). **e** Representative images of WT and *Npc1*<sup>−/−</sup> microglia plated onto APPPS1 cryosection. Microglial lysosomes were visualized with an antibody against CD68 (green), while Aβ plaques were detected with Thiazine Red (ThR) that stains fibrillar Aβ (red). Hoechst was used for nuclear staining (blue). Scale bar: 25 μm. **f** Quantification of Aβ plaque clearance was performed by comparing ThR positive area between a brain section incubated with microglia and the consecutive brain section where no cells were added. Values are expressed as percentages of amyloid plaque clearance normalized to the WT and represent mean ± SEM from three independent experiments ($n = 3$, $p = 0.039$, unpaired two-tailed Student's *t*-test).

TYROBP, APOE, ITGAX/CD11c and many late endosomal/lysosomal proteins, including LAMP1/2, LIPA, CD68, CTSB, CTSD or GRN. Among those, ITGAX showed the largest increase (39-fold). Concomitantly, markers associated with the homeostatic microglial function like P2RY12, TMEM119, CSF1R, CX3CR1, TGFB1 or TGFBR1 were decreased (Fig. 1b, Supplementary Data 1). Notably, Ingenuity Pathway Analysis (IPA) of MS data revealed phagosome maturation and autophagy as the two pathways mostly affected by the loss of NPC1 (Fig. 1c, Supplementary Fig. 2a), implicating a functional role of NPC1 in regulation of intracellular trafficking in microglia. Moreover, we compared our proteomic signatures with published transcriptome data of *Npc1*<sup>−/−</sup> microglia[50] and found a good correlation of DAM and homeostatic signatures, suggesting their transcriptional mode of regulation (Supplementary Fig. 2b). Western blot analysis of selected DAM signature proteins (Fig. 1d) confirmed our MS data. Of note, increased levels of lysosomal proteins such as CTSB or CTSD in *Npc1*<sup>−/−</sup> microglia were not accompanied by their reduced maturation as visualized by the increased levels of pro, single and heavy chain forms, suggesting that lysosomal degradation function, necessary for cathepsin maturation, is preserved upon loss of NPC1 (Fig. 1d).

To study the functional consequences of altered microglial proteomic signatures, we used an ex vivo amyloid β (Aβ) plaque clearance assay[64–66]. To this end, microglial phagocytic clearance of Aβ plaques was determined by measuring the reduction of fibrillar Aβ load visualized by Thiazine Red (ThR) staining after incubation of amyloid plaque bearing brain section with exogenously added microglia (Fig. 1e). Microglia from 8-weeks old *Npc1*<sup>−/−</sup> mice showed a 1.6-fold higher phagocytic clearance of amyloid aggregates compared to WT cells (Fig. 1f), suggesting that lysosomal degradation in *Npc1*<sup>−/−</sup> microglia is functional.

Taken together, our data show that a prominent neuroinflammation in *Npc1*<sup>−/−</sup> symptomatic mice is molecularly characterized by altered microglial proteomic signatures of intracellular trafficking, lysosomal function, and phagocytosis and functionally reflected by increased phagocytic clearance.

**Early changes in microglial *Npc1*<sup>−/−</sup> proteomic signatures occur prior to neuronal death and correlate with increased phagocytosis.** Lack of NPC1 is likely to affect microglial

function already during developmental stages when microglia are critically required for clearance of neuronal debris, synaptic pruning, and successful myelination that are pre-requisites for proper neuronal connectivity. To this end, we analyzed microglial phenotype at pre-symptomatic stages of *Npc1*<sup>−/−</sup> mice (P7). In agreement with the previous studies[15,16,47], Calbindin immunostaining delineated an intact layer of Purkinje cells, whereas cerebellar CD68 immunoreactivity was already upregulated in *Npc1*<sup>−/−</sup> microglia when compared to WT (Fig. 2a). This result implicates that despite the preserved neuronal environment, cerebellar *Npc1*<sup>−/−</sup> microglia were already activated at pre-symptomatic stages. Importantly, we also detected higher CD68 levels in less affected *Npc1*<sup>−/−</sup> regions, including cortex (Fig. 2b), similarly as shown for the symptomatic stages (Supplementary Fig. 1b and c).

To pinpoint early microglial molecular alterations, we analyzed the proteome of cerebral microglia from pre-symptomatic *Npc1*<sup>−/−</sup> animals. Although MS analysis of acutely isolated microglia from P7 *Npc1*<sup>−/−</sup> mice showed less pronounced changes (Fig. 2c, Supplementary Data 2) compared to the 8-weeks old *Npc1*<sup>−/−</sup> microglia (Fig. 1b, Supplementary Data 1), most of the late endosomal/lysosomal markers, including LAMP1/2, LIPA, CD63, CD68, CTSB, CTSD or GRN were already significantly upregulated. Upregulation of late endosomal/lysosomal proteins endorses microglial state switch as an early event in the NPC neuropathological cascade. Similar to the symptomatic stage, canonical pathway analysis of microglia from pre-symptomatic stages highlighted autophagy and phagosome maturation as the two most significantly altered pathways (Fig. 2d, Supplementary Fig. 2c). Moreover, the prediction of upstream regulator activity revealed signaling aberrations in *Npc1*<sup>−/−</sup> microglia, including the most prominent downregulation or upregulation in pathways regulated by TGFB1 and TFEB, respectively (Supplementary Fig. 2d). Further analysis of proteins regulated by TGFB1 is pinpointing alterations of endosomal/lysosomal signaling and reduced microglial homeostasis in *Npc1*<sup>−/−</sup> animals at both pathological stages (Supplementary Fig. 2e). Selected proteins found upregulated in the proteomic analysis of pre-symptomatic *Npc1*<sup>−/−</sup> mice were confirmed via western blot analysis (Fig. 2e). One of the most significantly changed proteins at the pre-symptomatic stage was the late endosomal and exosomal marker

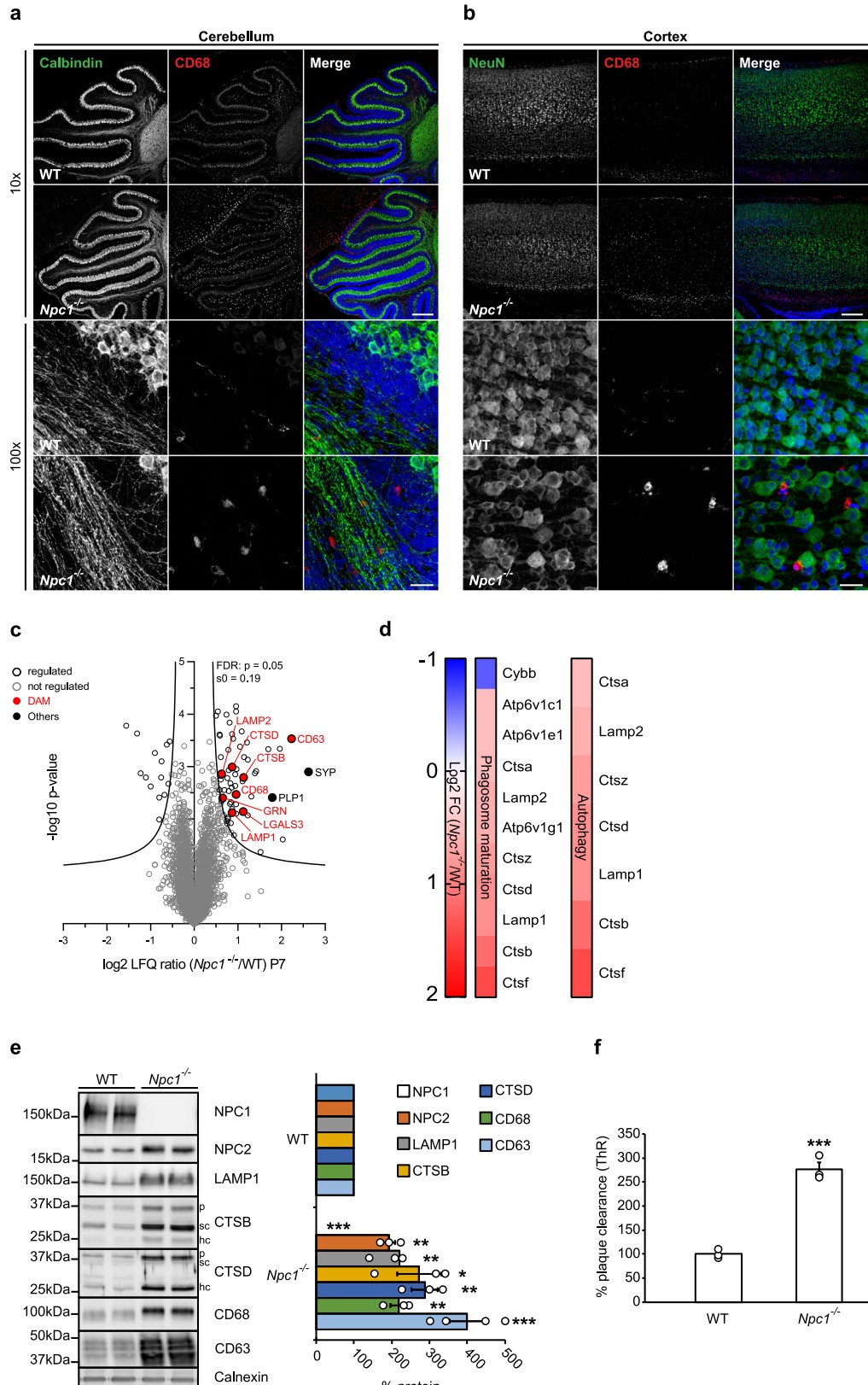

CD63 (Fig. 2c and e), which suggests that defects in endosomal/lysosomal trafficking and sorting may be among the earliest pathological alterations in $Npc1^{-/-}$ microglia[67,68]. Of note, CD63 was also increased at symptomatic stage, but as it could only be detected in $Npc1^{-/-}$ microglia it is not listed in the Supplementary Data 1.

Next, we tested whether proteomic fingerprint alterations were accompanied by functional changes in microglia. To this end, we performed the ex vivo Aβ plaque clearance assay. Microglia from pre-symptomatic $Npc1^{-/-}$ mice showed more than 2.5-fold increased phagocytic clearance compared to WT microglia (Fig. 2f). To further confirm that lysosomal function is preserved

**Fig. 2 Switch in microglial proteomic signatures and increased phagocytosis occur already in pre-symptomatic *Npc1*$^{-/-}$ mice and precede neuronal loss. a, b** Immunostaining of cerebellum (**a**) and cortex (**b**) of P7 WT and *Npc1*$^{-/-}$ mice from three independent experiments ($n = 3$) using antibodies against neuronal markers (green) Calbindin (Purkinje cells, cerebellum), NeuN (cortex) and lysosomal microglial marker CD68 (red) reveal no neurodegeneration in *Npc1*$^{-/-}$ mice (10×, upper panels). In contrast, CD68 immunoreactivity showed activated *Npc1*$^{-/-}$ microglia with amoeboid morphology already at this early pathological stage (100×, lower panels). Hoechst was used for nuclear staining (blue). Scale bars: 250 μm (10×, upper panels) and 25 μm (100×, lower panels). **c** Proteome analysis of acutely isolated microglia from P7 WT and *Npc1*$^{-/-}$ littermates. The negative log10 transformed *p*-value of each protein is plotted against its average log2 transformed LFQ ratio between *Npc1*$^{-/-}$ and WT microglia. The hyperbolic curves indicate the threshold for a permutation-based FDR correction for multiple hypotheses. Significantly changed proteins (*p*-value less than 0.05 and FDR corrected) with a log2 fold change larger than 0.5, or smaller than −0.5 are encircled in black (regulated) and not regulated proteins are encircled in gray. Selected DAM proteins (upregulated) are highlighted in red and the synaptic protein synaptophysin (SYP) and myelin protein (PLP1) in black (others). Microglia were analyzed from three independent experiments ($n = 3$). **d** *Npc1* deletion affects phagosome maturation and autophagy already at pre-symptomatic phase. The heatmap shows the average log2 transformed LFQ ratio between *Npc1*$^{-/-}$ and WT microglia of all significantly regulated proteins involved in phagosome maturation and autophagy according to IPA. **e** Validation of MS results via western blot analysis and corresponding quantification. Representative immunoblots of acutely isolated microglia from WT mice and *Npc1*$^{-/-}$ littermates showing increased level of late endosomal/lysosomal markers (NPC2, LAMP1, CTSB, CTSD, CD68, and CD63). Calnexin was used as a loading control. For cathepsins: p = pro-form; sc = single chain form; hc = heavy chain form. Proteins were quantified by densitometry (ImageJ—NIH) from at least three independent experiments. Values were normalized on WT control and represent mean ± SEM (unpaired two-tailed Student's *t*-test). NPC1 ($n = 4$, $p = 3 \times 10^{-6}$); NPC2 ($n = 3$, $p = 0.004$); LAMP1 ($n = 3$, $p = 0.006$); CTSB ($n = 3$, $p = 0.04$); CTSD ($n = 3$, $p = 0.006$); CD68 ($n = 3$, $p = 0.0058$); CD63 ($n = 4$, $p = 6 \times 10^{-4}$). **f** *Npc1*$^{-/-}$ microglia from P7 mice already show increased phagocytic capacity toward Aβ. Quantification of Aβ plaque clearance was performed by comparing ThR positive area between a brain section incubated with microglia and the consecutive brain section where no cells were added. Values are expressed as percentages of amyloid plaque clearance normalized to the WT and represent mean ± SEM from three independent experiments ($n = 3$, $p = 0.0003$, unpaired two-tailed Student's *t*-test).

in pre-symptomatic *Npc1*$^{-/-}$ microglia, we performed an in vitro pulse/chase epidermal growth factor receptor (EGFR) degradation assay. Western blot analysis and its densitometry quantification (Supplementary Fig. 3a and b) revealed comparable EGFR degradation rate in *Npc1*$^{-/-}$ and WT microglia, confirming that lysosomal degradation in microglia is not impaired by *Npc1* deletion. Thus, our data demonstrate that molecular and functional alterations of microglia are among the earliest pathological hallmarks in NPC that occur independently from the neuronal loss observed at later stages.

**Cell autonomous function of NPC1 in microglia.** To demonstrate a cell autonomous mechanism for microglial activation in NPC, we crossed a mouse line with a floxed allele of the *Npc1* gene (C57BL/6-Npc1tm1.2Apl)[69] with a mouse line expressing Cre recombinase under myeloid specific *Cx3cr1* promoter (B6. Cx3cr1tm1.1(cre)Jung/N)[70], thus generating a specific depletion of NPC1 in microglia in double transgenic mice (*Npc1*$^{flox/cre+}$). Correspondingly, microglia enriched fraction isolated from 8-weeks old *Npc1*$^{flox/cre+}$ mice showed a complete loss of NPC1 and a pronounced increase of late endosomal/lysosomal markers including LAMP1, CD68, and CD63 when compared to microglia enriched fraction of control mice (*Npc1*$^{flox/cre-}$) (Fig. 3a). These changes, triggered by the loss of NPC1 only in microglia, strongly support its role in late endosomal/lysosomal pathway and switch toward DAM signature that we described using the global *Npc1*$^{-/-}$ model. Accordingly, histological analysis showed that, despite an absence of neurodegeneration in all brain areas, including Purkinje cell layer, CD68 reactivity was significantly increased in cerebellum and cortex of *Npc1*$^{flox/cre+}$ compared to *Npc1*$^{flox/cre-}$ mice (Fig. 3b and c). Importantly, proteomic analysis of acutely isolated microglia from 5 months old *Npc1*$^{flox/cre+}$ mice revealed a fully established DAM signature (Fig. 3d, Supplementary Data 3), strongly resembling disease signatures we observed in symptomatic *Npc1*$^{-/-}$ mice (Supplementary Fig. 3c). These data confirm that NPC1 has a cell autonomous role in microglia and that chronic inflammation which is triggered by its loss is a key feature of NPC neuropathology that should be considered for therapeutic targeting.

*Npc1*$^{-/-}$ **microglia display enhanced uptake but impaired turnover of myelin**. It has been shown that *Npc1*$^{-/-}$ mice exhibit reduced myelination[71,72] that has been linked to impairments in oligodendrocyte development[33,34,73,74]. However, it is well appreciated that microglia engulf and clear myelin debris, regulating thereby myelin turnover[75]. This process can be impaired by cholesterol accumulation in microglia[76]. To compare myelin levels in the *Npc1*$^{-/-}$ and WT brain sections, we used a co-staining for a myelin protein (2′,3′-cyclic-nucleotide 3′-phosphodiesterase, CNPase) and compact myelin (Fluoromyelin). Symptomatic *Npc1*$^{-/-}$ mice showed reduced levels of both CNPase and Fluoromyelin compared to WT (Fig. 4a). Next, we analyzed removal of myelin debris by *Npc1*$^{-/-}$ microglia. We found that in cerebellum (Fig. 4b) and cortex (Fig. 4c) of symptomatic *Npc1*$^{-/-}$ mice almost all CD68 positive cells accumulated Fluoromyelin intracellularly. Similarly, myelin accumulation was also detected within microglia of symptomatic *Npc1*$^{nmf164}$ mice[52]. Noteworthy, we also observed colocalization between CD68 and myelin marker CNPase in *Npc1*$^{flox/cre+}$ mouse brain in contrast to *Npc1*$^{flox/cre-}$ control (Fig. 4d), suggesting that loss of NPC1 in microglia is sufficient to trigger increased myelin uptake. Importantly, our proteomic analysis identified increased levels of myelin-specific proteins (e.g., proteolipid protein 1, PLP1) and proteins involved in microglial phagocytosis of myelin (e.g., galectin 3/LGALS3) in both *Npc1*$^{-/-}$ (Fig. 2c, Supplementary Data 1 and 2) and *Npc1*$^{flox/cre+}$ microglia (Fig. 3d, Supplementary Data 3). Increased levels of PLP1 in microglia were also confirmed by western blot analysis (Fig. 4e, Supplementary Fig. 4a). Taken together, our analysis suggests that loss of NPC1 triggers hyper-reactive microglia that accumulate myelin and display prominent alterations of their proteomic signatures.

Increased levels of myelin proteins within *Npc1*$^{-/-}$ and *Npc1*$^{flox/cre+}$ microglia could reflect a higher uptake, but also possible defects in myelin turnover, or both. To address this question, we explored an ex vivo assay of myelin clearance and observed higher uptake of myelin in microglia from *Npc1*$^{-/-}$ pre-symptomatic mice, as visualized by the increased levels of Fluoromyelin within CD68 positive cells compared to WT controls (Supplementary Fig. 4b). Of note, increased myelin uptake phenotype by microglia is in line with increased uptake of

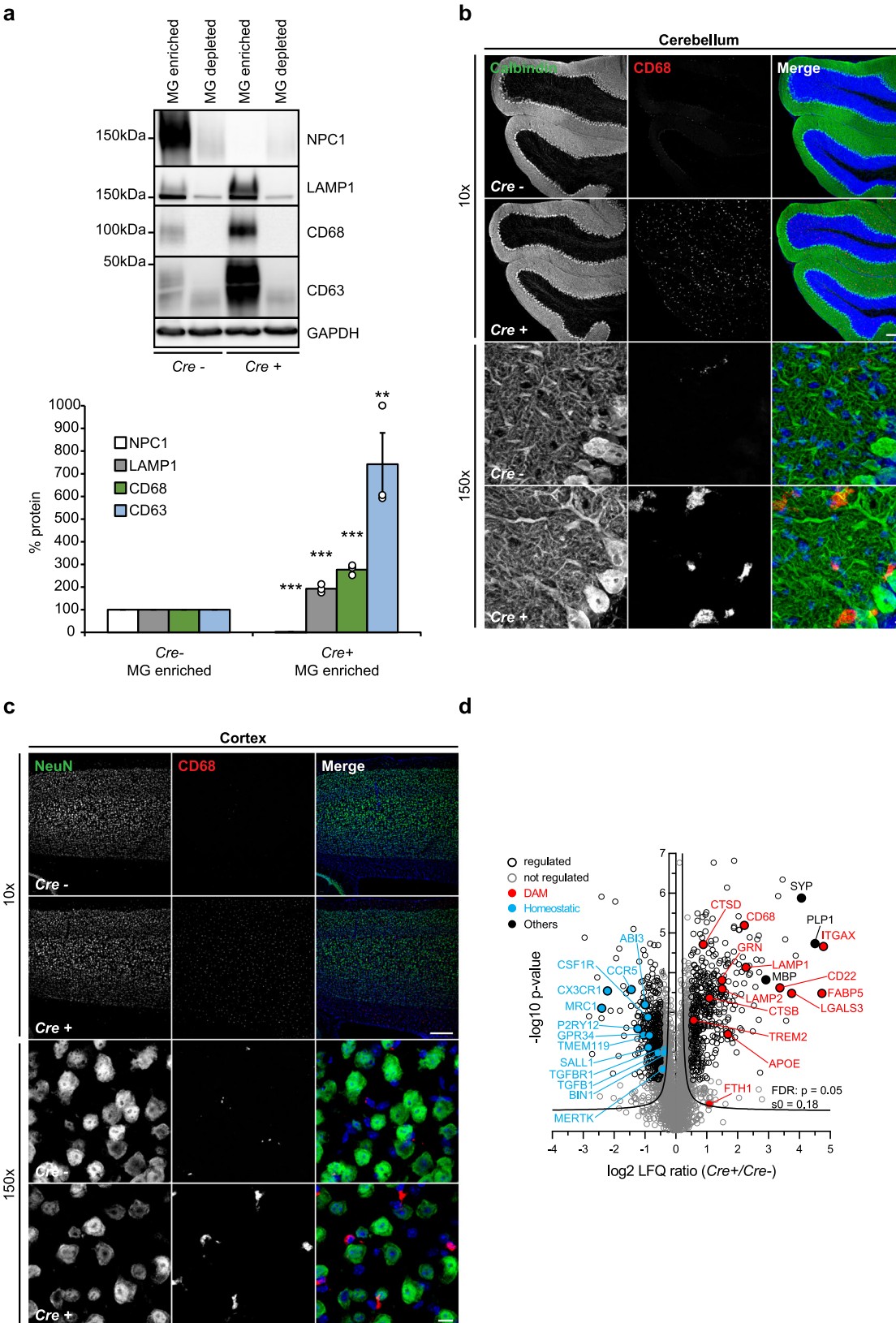

other substrates upon loss of NPC1, including Aβ plaques (Figs. 1f, 2f), synaptic proteins like synaptophysin (SYP) (Figs. 2c, 3d, Supplementary Data 1, 2, and 3) or Hoechst positive nuclear material (Supplementary Fig. 4c and d). Taken together, we observed increased phagocytic uptake of various microglial substrates upon loss of NPC1. In contrast to Aβ plaques that were efficiently degraded, myelin accumulated within $Npc1^{-/-}$ microglia (Supplementary Fig. 4b), suggesting possible impairments in myelin turnover.

Next, we investigated the turnover of exogenously added myelin in cultured $Npc1^{-/-}$ microglia isolated from pre-symptomatic mice. Cultured microglia have been pulsed for 6 h

**Fig. 3 Cell autonomous function of NPC1 in microglia. a** Western blot analysis and corresponding quantification of 8-weeks old $Npc1^{flox/cre+}$ ($Cre+$) mice with specific depletion of NPC1 in microglia and their littermate controls $Npc1^{flox/cre-}$ ($Cre-$). Representative immunoblots showing specific NPC1 depletion in microglia enriched (MG enriched) compared to microglia depleted (MG depleted) fraction of $Cre+$ microglia. Increased levels of late endosomal/lysosomal markers (LAMP1, CD68, and CD63) could only be detected in MG enriched fraction of $Cre+$ microglia. GAPDH was used as a loading control. Microglia enriched fraction from three independent experiments was quantified by densitometry (ImageJ—NIH). Values were normalized on $Cre-$ control and represent mean ± SEM (unpaired two-tailed Student's $t$-test). NPC1 ($n = 3$, $p = 6 \times 10^{-10}$); LAMP1 ($n = 3$, $p = 3 \times 10^{-4}$); CD68 ($n = 3$, $p = 2 \times 10^{-4}$); CD63 ($n = 3$, $p = 0.0097$). **b**, **c** Immunostaining of cerebellum (**b**) and cortex (**c**) from 8-weeks old $Npc1^{flox/cre+}$ ($Cre+$) and $Npc1^{flox/cre-}$ ($Cre-$) mice from three independent experiments ($n = 3$) using antibodies against neuronal markers (green) Calbindin (Purkinje cells, cerebellum) and NeuN (cortex) and lysosomal microglial marker CD68 (red). No neurodegeneration was observed in any of the investigated brain regions of $Cre+$ mice (×10, upper panels). In contrast, increased CD68 immunoreactivity and amoeboid morphology were observed (×150, lower panels). Hoechst was used for nuclear staining (blue). Scale bars: 250 μm (×10, upper panels) and 10 μm (×150, lower panels). **d** Proteome analysis of acutely isolated microglia from 5 months old $Npc1^{flox/cre+}$ ($Cre+$) and $Npc1^{flox/cre-}$ ($Cre-$) mice. The negative log10 transformed $p$-value of each protein is plotted against its average log2 transformed LFQ ratio between $Cre+$ and $Cre-$ microglia. The hyperbolic curves indicate the threshold for a permutation-based FDR correction for multiple hypotheses. Significantly changed proteins ($p$-value less than 0.05 and FDR corrected) with a log2 fold change larger than 0.5, or smaller than −0.5 are encircled in black (regulated) and not regulated proteins are encircled in gray. Selected homeostatic (downregulated) and DAM (upregulated) proteins are in addition highlighted in blue and in red, respectively. Synaptic protein SYP and myelin proteins MBP and PLP1 are highlighted in black (others). Microglia were analyzed from three independent experiments ($n = 3$).

with purified and fluorescently labeled myelin[76], and myelin turnover was monitored over 72 h (Fig. 5a). After 6 h, myelin was mainly found inside CD68 positive late endosomal/lysosomal compartments in both WT and $Npc1^{-/-}$ cells. At 48 h, most of the fluorescent labeling in WT cells was confined within vesicles outside of CD68 positive late endosomal/lysosomal compartments (Fig. 5b). Immunostaining for the membrane marker Perilipin 2 confirmed that these vesicles were lipid droplets, cellular organelles specialized for lipid recycling and storage and crucial for cell metabolism[77] (Supplementary Fig. 5). In contrast to WT cells, $Npc1^{-/-}$ microglia accumulated labeled myelin within the late endosomal/lysosomal compartment, suggesting an impairment in myelin turnover and recycling into lipid droplets (Fig. 5b). Even after 72 h, $Npc1^{-/-}$ microglia still contained the engulfed myelin within late endosomal/lysosomal compartment and completely lacked fluorescently labeled lipid droplets (Fig. 5a), demonstrating a severe impairment and not just a delay in the myelin degradation process. To further confirm microglial impairment in myelin processing in $Npc1^{-/-}$ microglia, cells were analyzed for lipid droplet formation at 72 h using the lipophilic dye Nile red[78] (Fig. 5c). Imaging analysis confirmed the presence of Nile red positive lipid droplets localized to the cell periphery in WT microglia. Again, in $Npc1^{-/-}$ microglia, lipid droplet staining could not be observed and Nile red visualized accumulated myelin within late endosomes/lysosomes.

**Myelin accumulates in late endosomes/multivesicular bodies in $Npc1^{-/-}$ mice.** To further characterize endosomal/lysosomal defects, we performed an electron microscopy (EM) analysis of acutely isolated microglia from pre-symptomatic $Npc1^{-/-}$ and control WT mice. This analysis revealed pronounced accumulation of late endosomes/multivesicular bodies (MVBs) within $Npc1^{-/-}$ microglia (Fig. 6a and b (1)). Morphological analysis showed that most of these vesicles contained undigested lipid material (Fig. 6a and b (2)), further supporting that $Npc1^{-/-}$ microglia hold an increased phagocytic capacity toward myelin, but fail in its processing. Interestingly, in contrast to increased numbers of MVBs, no obvious increase in numbers of lysosomes (Fig. 6a) were observed in microglia by EM as it has been previously suggested in other brain cells[23]. This is in agreement with preserved lysosomal degradation function and suggests that myelin accumulates in late endosomes/MVBs and thus does not reach lysosomes where it is normally processed. To demonstrate a trafficking defect and myelin accumulation in late endosomes/MVBs, rather than lysosomes, we performed a pulse/chase experiment with primary microglia from pre-symptomatic

$Npc1^{-/-}$ mice (Fig. 6c). Microglial cells were fed with fluorescently labeled myelin for 15 min, and myelin trafficking was followed over 1 h using fluorescently labeled CTX to visualize endosomes and DQ-BSA to label lysosomes. Confocal microscopy analysis revealed that in WT cells most of the phagocytosed myelin reached lysosomal compartment after 1 h. In contrast, myelin was poorly colocalizing with DQ-BSA in $Npc1^{-/-}$ microglia (Fig. 6d). Taken together, despite the enhanced phagocytic uptake by $Npc1^{-/-}$ microglia, lipid trafficking toward the lysosomes was impaired, thus resulting in myelin accumulation in late endosomes/MVBs and defects in lipid clearance.

**Lowering cholesterol rescues lipid droplet formation and reduces pathological signatures of $Npc1^{-/-}$ microglia.** Many different lipid alterations have been reported in NPC and it is debated which lipid is causative for neuropathological defects[10]. To gain mechanistic insights into lipid alterations that are causative for microglial phenotypes we identified, we first lowered cholesterol due to the major regulatory effect of NPC1 on cholesterol transport[79]. Addition of cholesterol-lowering drug methyl-β-cyclodextrin (MβCD) partially restored lipid droplet formation in $Npc1^{-/-}$ microglia (Fig. 7a). As already shown upon in vivo treatment of $Npc1^{-/-}$ mice, MβCD is able to alter microglial transcriptional signatures[50], suggesting that it may reverse excessive DAM response of $Npc1^{-/-}$ microglia. Thus, we looked for the rescue of microglial homeostatic signatures by analyzing levels of the homeostatic marker TGFB1[80] and DAM marker CTSD in cultured microglia upon MβCD treatment. MβCD treatment of $Npc1^{-/-}$ microglia associated with a pronounced increase in levels of TGFB1 and, concomitantly, decreased levels of CTSD, supporting its potential protective effect on enhancing microglial homeostasis and reducing DAM signatures (Fig. 7b). Of note, as already reported, cultured microglia show altered transcriptomic signatures with reduced homeostatic and increased inflammatory profiles[80,81], mitigating differences between $Npc1^{-/-}$ and WT microglia in the absence of an additional challenge. This underscores the importance of proteomic studies in acutely isolated microglia. Taken together, these experiments suggest that cholesterol accumulation in microglia may, at least partially, be responsible for microglial functional deficits and suggest the beneficial effect of cholesterol-lowering drugs on reducing microglial activation.

**Macrophages from NPC patients feature murine NPC1-deficient microglial phenotypes.** $Npc1^{-/-}$ and $Npc1^{flox/cre+}$

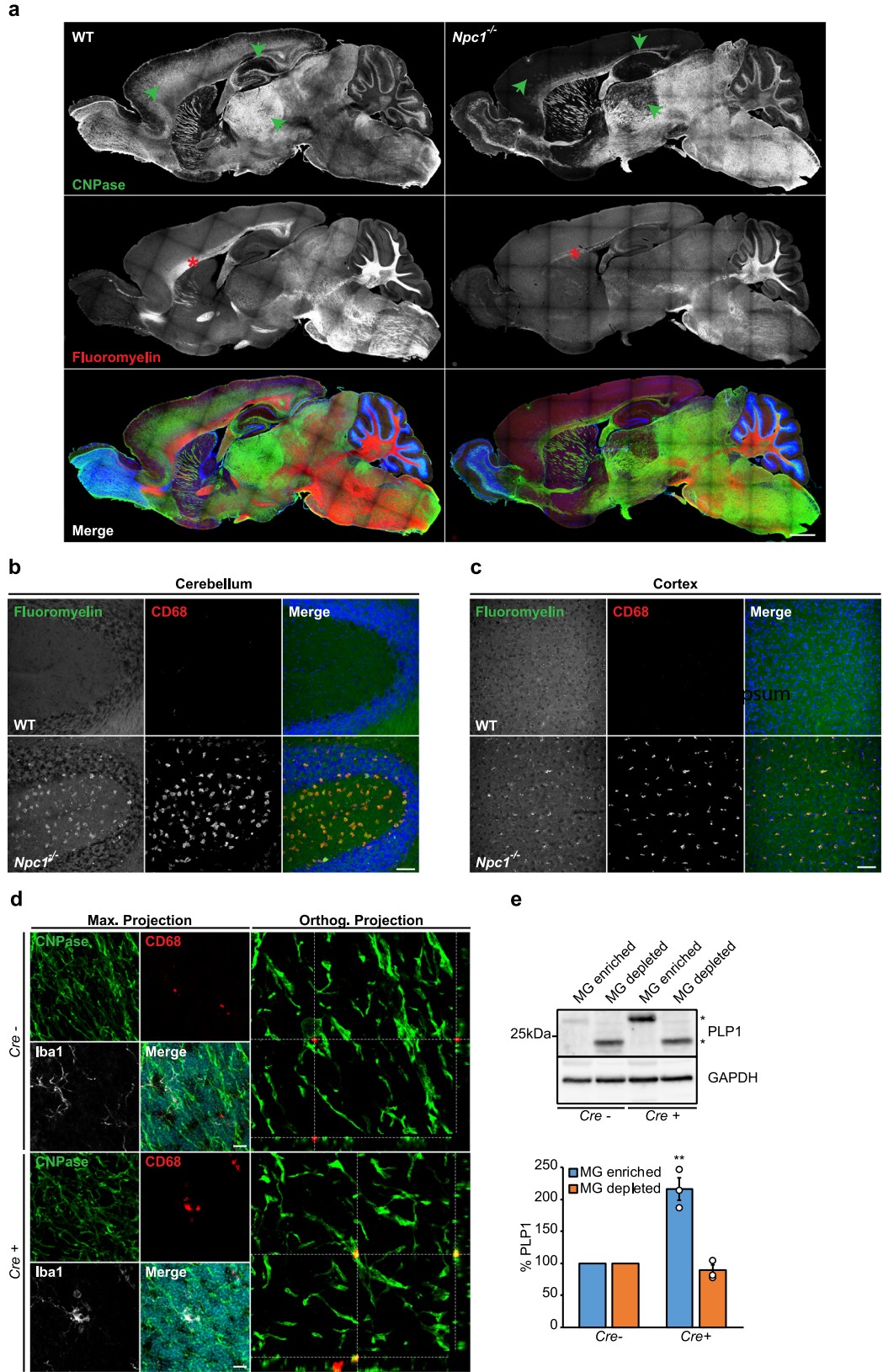

microglia offer potential for therapeutic intervention and bio-marker studies, but translation of animal models to human patients is limited due to the very small number of NPC patients (estimated at 1:100,000 live births) and limited access to viable human brain tissue[82]. As a compromise, we explored human peripheral macrophages for their phenotypic alterations in NPC.

First, we performed a proteomic analysis of peripheral blood monocyte-derived macrophages from 7 NPC patients and 3 healthy controls (Fig. 8a, Supplementary Data 4). Despite the larger variability among the human samples, this analysis revealed a tendency of increased levels of late endosomal/lyso-somal and DAM proteins, including LAMP1/2, NPC2, ITGAX,

**Fig. 4 Microglial NPC1 loss of function triggers increased myelin uptake. a** Tile scan of whole brain section from 8-weeks old WT and $Npc1^{-/-}$ mice from three independent experiments ($n = 3$) immunostained using antibody against CNPase as a general myelin marker (green) and Fluoromyelin that visualizes compact myelin (red). Hoechst was used for nuclear staining (blue). Significantly reduced myelin levels are observed in $Npc1^{-/-}$ cortex and hippocampus as shown by reduced CNPase immunoreactivity (green arrowheads). Fluoromyelin also reveals reduced levels of myelin in $Npc1^{-/-}$ mouse compared to WT control (red stars). Scale bar: 750 μm. **b**, **c** Microglia engulf myelin in symptomatic $Npc1^{-/-}$ mice. Immunohistological analysis demonstrates Fluoromyelin (green) positive myelin debris inside of CD68 positive (red) late endosomal/lysosomal compartments in cerebellum (**b**) and cortex (**c**) of 8-weeks old $Npc1^{-/-}$ compared to WT mice from three independent experiments ($n = 3$). Scale bars: 50 μm. **d** Immunohistological analysis showing myelin marker CNPase (green) colocalizing with lysosomal marker CD68 (red) of Iba1 positive microglia (white) in cerebellum of 8-weeks old $Npc1^{flox/cre+}$ ($Cre+$) compared to $Npc1^{flox/cre-}$ ($Cre-$) mice from three independent experiments ($n = 3$). Hoechst was used for nuclear staining (blue). Scale bars: 10 μm. **e** Western blot analysis and corresponding quantification of myelin-specific protein PLP1 in 8-weeks old $Npc1^{flox/cre+}$ ($Cre+$) mice and their littermate controls $Npc1^{flox/cre-}$ ($Cre-$). Representative immunoblots from $Cre+$ and $Cre-$ mice showing PLP1 accumulation (*) occurring in MG enriched fraction of $Cre+$ mice. Of note, PLP1 running behavior differs between MG enriched and MG depleted fraction and higher molecular weight of PLP1 can be detected in MG enriched fraction. This effect may be attributed to the aggregation of myelin proteins reported in microglia[75]. GAPDH was used as a loading control. PLP1 levels were quantified by densitometry (ImageJ—NIH) from three independent experiments. Values were normalized on $Cre-$ control and represent mean ± SEM ($n = 3$, $p = 0.003$, unpaired two-tailed Student's $t$-test).

CTSD, CTSB, APOE, GRN, and LIPA in NPC patient macrophages compared to healthy controls (Fig. 8a). Furthermore, increased levels of NPC2, LAMP1, CTSD, CD68, and GRN could be validated by western blot analysis (Fig. 8b). Thus, analysis of mouse microglia and NPC patient macrophages shows partially overlapping alterations in proteomic signatures upon loss of NPC1 and reveals its critical role in regulating endosomal/lysosomal homeostasis of myeloid cells. Noteworthy, in contrast to rodent $Npc1^{-/-}$ microglia that fully lack NPC1 (Fig. 1d), patient macrophages have residual levels of NPC1 (Fig. 8b) that may contribute to less prominent alterations in proteomic signatures.

Next, we functionally characterized patient macrophages using our ex vivo Aβ plaque clearance assay. Despite patient variability, NPC1-deficient human macrophages showed a tendency toward higher amyloid plaque clearance capacity in comparison to control cells, resembling phenotypic features of rodent $Npc1^{-/-}$ microglia (Fig. 8c and d). To further translate our observations from NPC1-deficient murine cells, we also performed the in vitro myelin phagocytic assay using patient cells (Fig. 8e). As we observed for the murine model, human NPC patient cells were capable to efficiently uptake exogenously added myelin, as judged by the intracellular fluorescent signal that could be detected in control and patient cells. At 48 h, fluorescently labeled lipid droplets were observed at the periphery in control cells. Similar to murine $Npc1^{-/-}$ microglia, fluorescently labeled lipid droplets could not be detected at the cell periphery of NPC patient cells, suggesting that a trafficking defect may preclude myelin degradation and recycling into lipid droplets.

In conclusion, our data show that NPC patient macrophages recapitulate many of the key molecular and phenotypic features of murine $Npc1^{-/-}$ microglia and thus represent a valuable tool to identify biomarkers and monitor disease progression and therapeutic interventions in NPC patients.

## Discussion

Our study reveals pathological proteomic fingerprints of microglia already in pre-symptomatic phase of NPC that associate with enhanced microglial phagocytic uptake and impairment in lipid trafficking, characterized by aberrant delivery of myelin into lysosomes and its accumulation within MVBs. This results in myelin turnover defects and severe impairment in lipid droplet formation that are accompanied by pronounced microglial inflammation and compromised function. Microglia are responsible for the neuroinflammatory phenotype that occurs in brains of $Npc1^{-/-}$ mice as infiltration of peripheral macrophages could not be detected in this model[50,53,83]. However, the contribution of microglial activation to neurodegeneration in NPC is still debated[28,47,52,55,57,58]. We detected pronounced microglial

reactivity throughout cortex and hippocampus prior to neuronal loss that complements previous reports[47,50]. Furthermore, our molecular and functional characterization reveals microglial dysfunction as an early pathological insult in NPC. Importantly, we demonstrate a cell autonomous role of NPC1 in microglia as loss of NPC1 function in microglia only is sufficient to trigger microglial activation and pathological proteomic signatures. Taken together, our data show that microglial alterations are early neuropathological hallmarks of NPC that should be considered for therapeutic targeting.

In contrast to in-depth studies of microglial transcriptomic profiles across neurodegenerative diseases, microglial proteomic signatures are less characterized. Recent transcriptomic studies revealed upregulation of DAM signatures in $Npc1^{-/-}$ mice[50,84] and overlap that we observe between transcriptomic and proteomic profiles suggests a transcriptional mode of regulation for the major DAM profiles in NPC, similarly as we described in AD microglia[63]. So far, it is unclear how DAM signatures across various neurodegenerative diseases reflect microglial function and whether they are predictive of beneficial or detrimental responses. Here, we demonstrate that DAM signatures correlate with increased phagocytic uptake both in murine $Npc1^{-/-}$ microglia and in human NPC macrophages. Similarly, in a LSD caused by granulin loss of function ($Grn^{-/-}$)[42,85], DAM signatures correlated with increased microglial phagocytosis[86]. In contrast, microglial cells in AD acquire DAM signatures[36,38], but impaired phagocytic function has been reported[59,87,88]. We recently delineated proteomic signatures of AD microglia and discovered Microglial Aβ Response Proteins (MARPs) that reflect microglial alterations triggered by Aβ accumulation[63]. Although NPC and AD microglia share many of the key MARPs (e.g., ITGAX, APOE, LGALS3, TREM2, CD68, and CD63), they differ in phagocytic capacity toward Aβ. Thus, our data strengthen the fundamental importance of combining molecular fingerprint analysis with functional studies to identify molecular targets that are predictive of microglial function and could be explored for repair. NPC patients typically do not show amyloid plaque deposition, but do have increased Aβ production[89–92]. This was ascribed as a consequence of the limited life expectancy of NPC patients, but it is also tempting to speculate that increased phagocytic Aβ clearance in NPC may compensate for the increased Aβ generation and thereby counteract amyloid deposition.

Our proteomic profiles of acutely isolated $Npc1^{-/-}$ microglia revealed significant upregulation of numerous late endosomal/lysosomal proteins (e.g., LAMP1/2, CD63, CD68, CTSB, and CTSD) already in pre-symptomatic mice. Moreover, we show that specific deletion of $Npc1$ in microglia is sufficient to

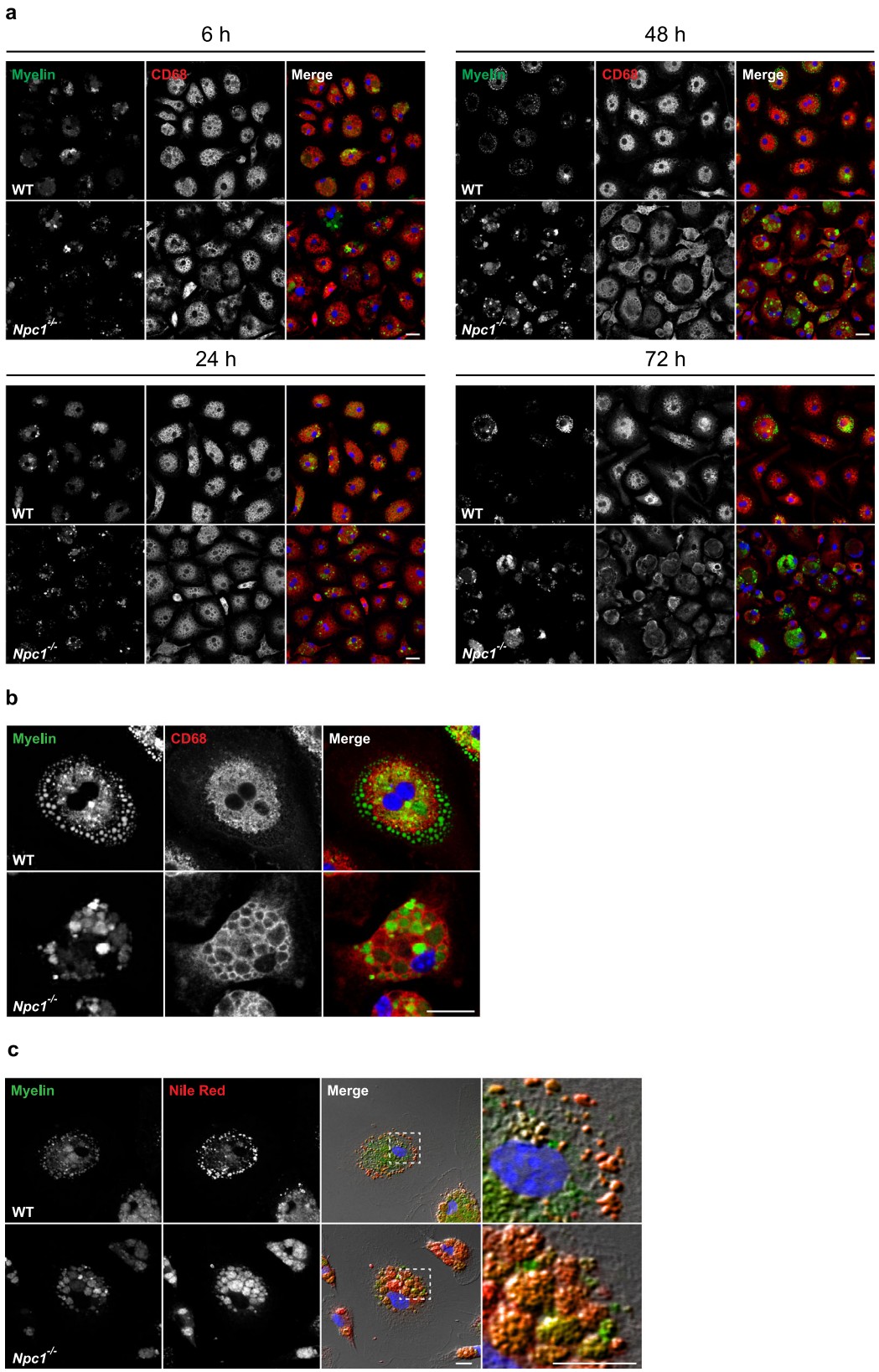

convert microglia toward a DAM phenotype. Thus, our study demonstrates that microglial activation is a direct consequence of loss of NPC1 function in microglia and not just an immune response mediated by degenerating brain environment. Importantly, our analysis of NPC patient macrophages revealed some of the changes observed in murine NPC1-deficient microglia (e.g.,

increased levels of LAMP1/2, CD68, CTSD, GRN, APOE or ITGAX), albeit to a lesser degree. This may be contributed by different degrees of NPC1 reduction (full loss of NPC1 in mouse model versus partial reduction in human cells), increased experimental variability or intrinsic differences between microglia and macrophages. Another limitation of our study is the

**Fig. 5 Npc1$^{-/-}$ microglia display impairment in myelin turnover. a, b** In vitro myelin phagocytosis assay. Cultured primary microglia isolated from P7 mice from three independent experiments (n = 3) were incubated with fluorescently labeled myelin (green) and analyzed at 24, 48, and 72 h. Microglial lysosomes were stained with anti-CD68 antibody (red). Hoechst was used for nuclear staining (blue). **a** Both WT and Npc1$^{-/-}$ microglia showed efficient myelin uptake at 6 h. At 24 h (lower left panels) fluorescent labeling could be found within CD68 positive late endosomal/lysosomal compartments in both WT and Npc1$^{-/-}$ microglia. At 48 h (upper right panels and higher magnification images shown in **b**), fluorescently labeled lipid vesicles were observed in WT microglia, indicating myelin turnover. In contrast, Npc1$^{-/-}$ microglia accumulated myelin within the CD68 positive compartments and no fluorescently labeled lipid vesicles were detected. At 72 h (lower right panels), in most of the WT cells myelin was degraded or signal could be detected in lipid vesicles. In Npc1$^{-/-}$ microglia, myelin signal was still within the CD68 positive compartments, suggesting compromised myelin turnover. Scale bars: 25 μm.
**c** Primary microglia isolated from P7 WT and Npc1$^{-/-}$ mice from three independent experiments (n = 3) were fed with fluorescently labeled myelin (green) and analyzed after 72 h. In order to confirm the identity of lipid vesicles forming as a result of myelin turnover, we stained microglia with Nile red (red) to visualize lipid droplets. Boxed regions are enlarged in right panels and reveal co-localization between fluorescently labeled myelin vesicles and Nile red, supporting myelin turnover and lipid droplet formation in WT microglia. In Npc1$^{-/-}$ microglia, Nile red mainly stained myelin deposits accumulating in late endosomes/lysosomes, instead of lipid droplets at the cell periphery, confirming impairment in myelin turnover. Scale bars: 10 μm.

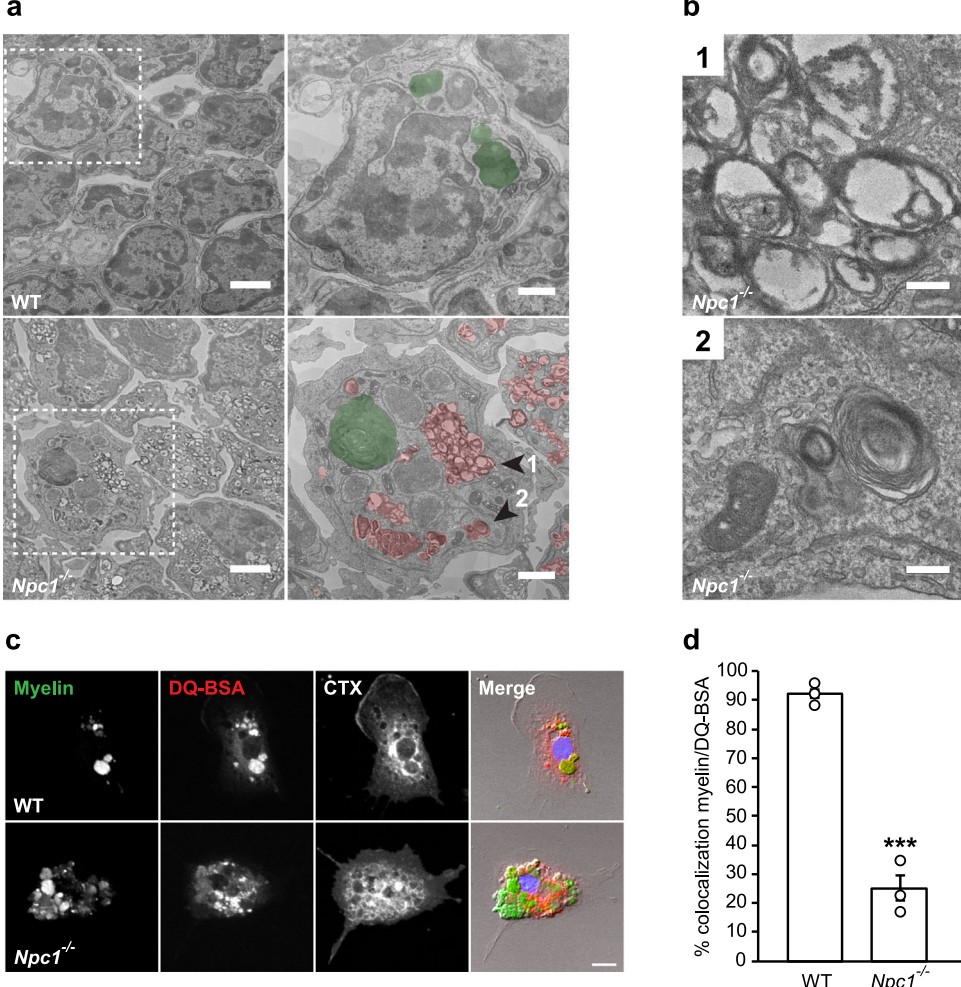

**Fig. 6 Myelin accumulation in late endosomes/MVBs of Npc1$^{-/-}$ microglia. a** EM analysis of microglia acutely isolated from pre-symptomatic WT and Npc1$^{-/-}$ mice from three independent experiments (n = 3) shows the significant intracytoplasmic accumulation of MVBs in Npc1$^{-/-}$ cells. Boxed regions are depicted at higher magnification in right panels (MVBs and lysosomes are pseudocolored in red and green, respectively). Scale bars: 2 μm (left panels) and 1 μm (right panels). **b** Representative high magnification images of Npc1$^{-/-}$ cell shown in (**a**) (arrowheads 1 and 2) revealing accumulation of MVBs (1). Undigested lipid material (lamellar structures) could be detected within MVBs (2). Scale bars: 0.2 μm. **c** Transport of phagocytosed myelin (green) was followed over 1 h in P7 WT and Npc1$^{-/-}$ microglia from three independent experiments (n = 3) using DQ-BSA to visualize lysosomal compartments (red) and fluorescently labeled cholera toxin (CTX, white) to visualize endocytic vesicles. Scale bar: 10 μm. **d** Quantification of the in vitro myelin transport assay shows that, in contrast to WT control, phagocytosed myelin did not reach lysosomal compartments in Npc1$^{-/-}$ microglia. Values are expressed as percentages of myelin signal colocalizing with DQ-BSA positive lysosomes and represent mean ± SEM from three independent experiments (n = 3, p = 0.0003, unpaired two-tailed Student's t-test).

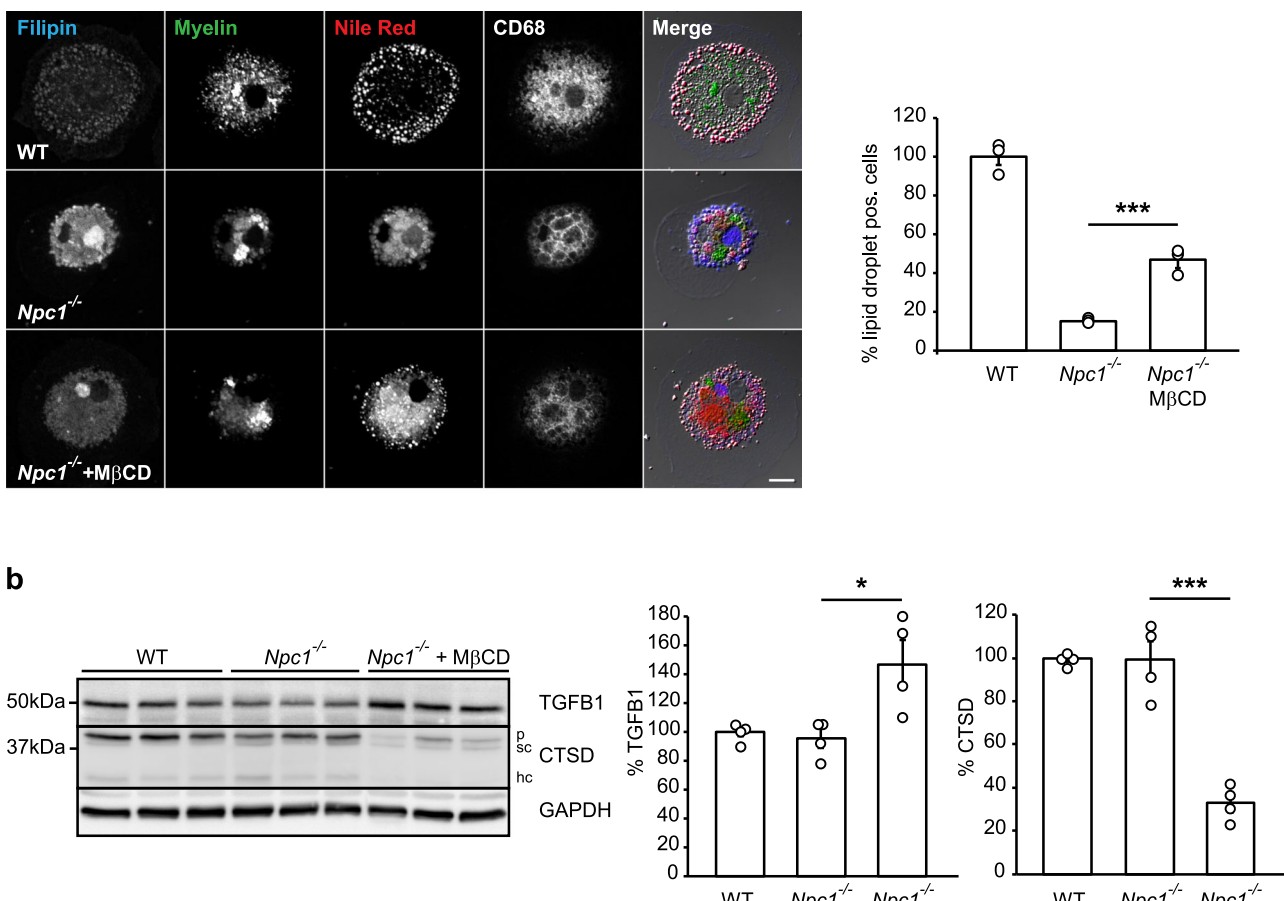

**Fig. 7 Lowering cholesterol in *Npc1*⁻/⁻ microglia rescues lipid droplet phenotype. a** MβCD treatment (48 h) of cultured *Npc1*⁻/⁻ microglia isolated from P7 mice and fed with fluorescently labeled myelin (green) partially restored the formation of lipid droplets as visualized by Nile red (red). Filipin dye (blue) and an anti-CD68 antibody (white) were used to visualize cholesterol and late endosomal/lysosomal compartments, respectively. Scale bar: 10 μm. Cells showing Nile red positive structures at cell periphery were quantified as lipid droplet positive cells. Three independent experiments were analyzed by confocal microscopy ($n = 3$, $p = 0.0003$, one-way ANOVA followed by unpaired two-tailed Student's $t$-test). **b** Western blot analysis and quantification of TGFB1 and CTSD (p = pro-form; sc = single chain form; hc = heavy chain form) in cultured *Npc1*⁻/⁻ and WT microglia isolated from P7 mice. At 5DIV, *Npc1*⁻/⁻ cells were treated with 250 μM MβCD for 48 h. GAPDH was used as a loading control. Protein levels were quantified by densitometry (ImageJ—NIH) from four independent experiments. Values were normalized on WT control and represent mean ± SEM (one-way ANOVA followed by unpaired two-tailed Student's $t$-test). TGFB1 ($n = 4$, $p = 0.02$); CTSD ($n = 4$, $p = 0.0002$).

treatment of recruited NPC patients with the approved drug miglustat[93]. Differences in the length of treatment period and patient-to-patient responsiveness could also contribute to reported proteomic profiles.

Noteworthy, and in line with our study, increased levels of cathepsins were reported in brain and plasma of NPC patients[56,94]. Along these lines, upregulation of pro-inflammatory gene expression in both murine and human brains, as well as inflammatory CSF signatures of NPC patients, suggest that immune changes are early pathological hallmarks of NPC[48]. Moreover, beside NPC, neuroinflammation is emerging as a common pathological denominator in other LSDs, suggesting that microglia are vulnerable to lysosomal impairments and that neuroinflammation contributes to neuronal death[95,96]. These findings are in line with our study showing that loss of NPC1 results in a strong pro-inflammatory phenotype and dysregulated microglia at early pre-symptomatic disease stages. Accordingly, pharmacological or genetic immune modulations have been shown to be beneficial and improved life expectancy in NPC murine models[17,50,54].

Uncontrolled and overt microglial activation during early *Npc1*⁻/⁻ development can compromise physiological processes and neuronal homeostasis. It is known that microglia play a role in synaptic pruning[97,98] and synaptic changes together with axonal pathology are reported in *Npc1*⁻/⁻ mice[57,99]. Of note, increased uptake of neuronal material that we detect in *Npc1*⁻/⁻ microglia is in line with the previous reports[50,100] and could impair neuronal connectivity as described in AD, schizophrenia and *Grn*⁻/⁻ models[101–104]. Psychiatric symptoms are often found in NPC patients, particularly in individuals with juvenile onset disease, including mental retardation, behavioral problems, schizophrenia-like psychosis, bipolar disorder, and attention deficit hyperactivity disorder[4,105]. Further work is needed to clarify whether excessive microglial activity we describe here may compromise synaptic pruning and contribute to psychiatric manifestations reported in NPC patients and this patho-mechanism may be of relevance for other psychiatric disorders.

In addition to synaptic pruning, microglia orchestrate myelin homeostasis, especially during development. Microglia control OPC recruitment and differentiation[106,107] and are responsible

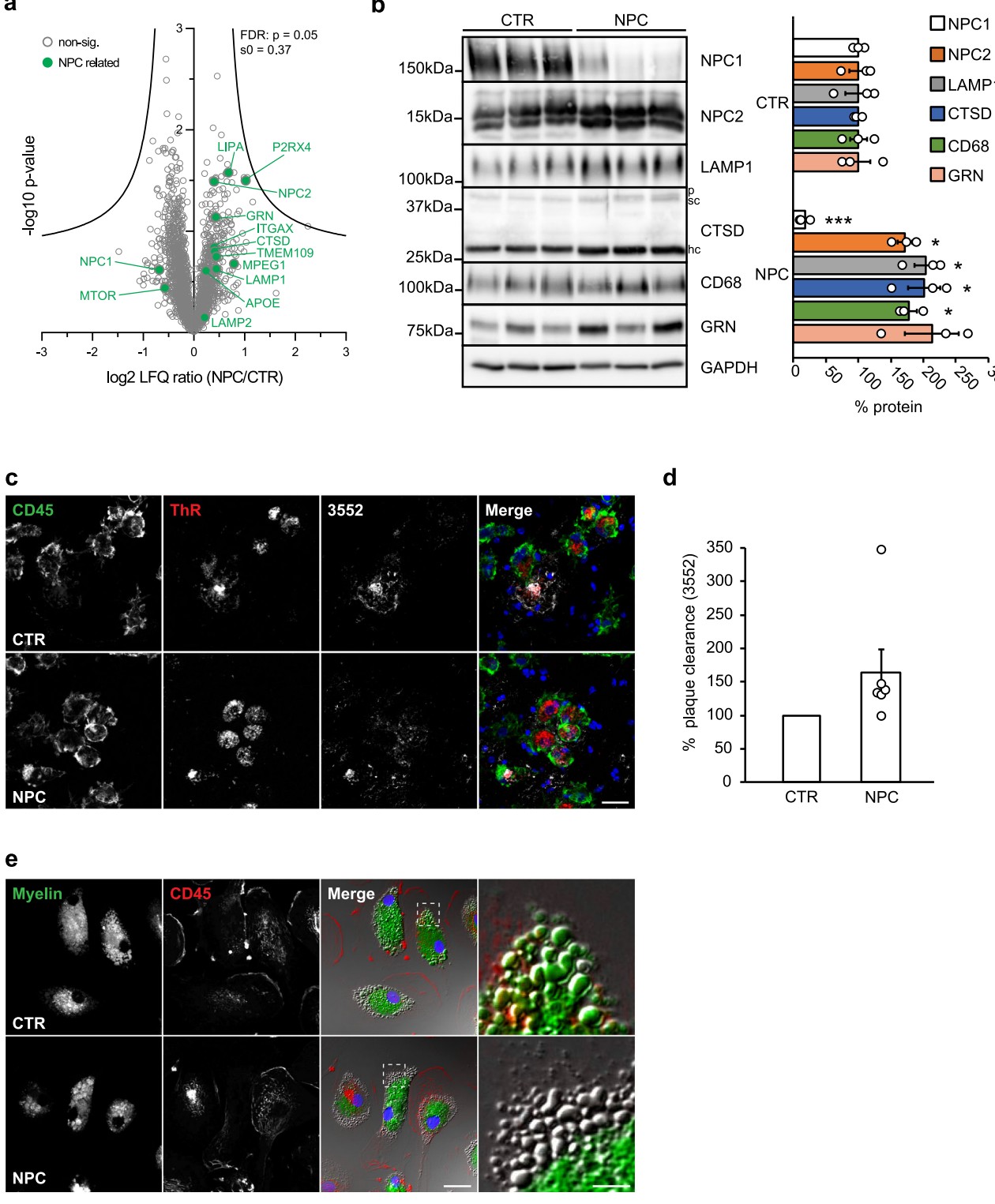

for the clearance of myelin debris. Impaired removal of myelin debris by microglia can heavily compromise re-myelination upon injury[52,76]. The hypomyelination phenotype described in NPC mouse models and human patients has been mainly associated to a cell autonomous impairment of oligodendrocytes[33,34,71–73,108]. Our data reveal that at pre-symptomatic stage, when myelination process just started[109], $Npc1^{-/-}$ microglia already show increased myelin uptake. Furthermore, LGALS3, that is expressed at high levels in microglia, triggers OPC differentiation and, together with TREM2, regulates myelin phagocytosis[110]. Our proteomic

dataset showed that LGALS3 abundance is strongly increased both in $Npc1^{-/-}$ and $Npc1^{flox/cre+}$ microglia, further supporting higher phagocytic capacity toward myelin. Accordingly, early accumulation of LGALS3 was found in blood serum of NPC patients, suggesting its potential as a biomarker for therapeutic monitoring[56]. Moreover, myelin accumulation and higher levels of several myelin proteins (e.g., PLP1 and MBP) were detected in $Npc1^{flox/cre+}$ microglia in vivo. Thus, our data suggest a cell autonomous defect in myelin turnover triggered by the loss of NPC1 in microglia. This collective evidence suggests that, in

**Fig. 8 Human blood-derived macrophages from NPC patients resemble pathological alterations of murine *Npc1*$^{-/-}$ microglia. a** Proteome analysis of macrophages from 7 NPC patients (NPC) and 3 healthy controls (CTR). The negative log10 transformed *p*-value of each protein is plotted against its average log2 transformed LFQ ratio between NPC and CTR macrophages. The hyperbolic curves indicate the threshold for a permutation-based FDR correction for multiple hypotheses. Selected proteins altered in human NPC macrophages are highlighted in green and non-significantly changed proteins are encircled in gray. **b** Validation of MS data via western blot analysis and corresponding quantification. Representative immunoblots from 3 NPC patients and 3 healthy controls reveal increased levels of late endosomal/lysosomal proteins in NPC patient-derived macrophages. GAPDH was used as loading control and quantification was performed by densitometry (ImageJ—NIH) from three independent experiments. Values were normalized on CTR and represent mean ± SEM (unpaired two-tailed Student's *t*-test). NPC1 (*n* = 3, *p* = 0.0003); NPC2 (*n* = 3, *p* = 0.02); LAMP1 (*n* = 3, *p* = 0.02); CTSD (*n* = 3, 0.01); CD68 (*n* = 3, *p* = 0.01); GRN (*n* = 3, *p* = 0.07). **c** Ex vivo Aβ plaque clearance assay. Representative immunostaining showing macrophages (CD45, green) plated onto APPPS1 brain section that are clustering around and phagocytosing Aβ plaques, visualized with Thiazine Red (ThR, plaque core, red) and 3552 anti-Aβ antibody (Aβ plaque, white). Hoechst was used for nuclear staining (blue). Scale bar: 25 µm. **d** Quantification of phagocytosed Aβ plaques. Values are expressed as percentages of amyloid plaque clearance normalized to CTR and represent mean ± SEM from 6 NPC patients and 3 healthy controls (*p* = 0.19, unpaired two-tailed Student's *t*-test). **e** In vitro myelin phagocytosis assay. Human macrophages from 2 NPC patients and 2 healthy controls were fed with fluorescently labeled myelin (green) and analyzed after 48 h. Myeloid cells were visualized using an antibody against CD45 (red). Hoechst was used as nuclear staining (blue). Boxed regions are enlarged in right panels and show that human CTR macrophages efficiently uptake and turnover myelin as demonstrated by fluorescently labeled lipid droplets at the cell periphery. In contrast, we could not detect fluorescently labeled lipid droplets at the cell periphery in NPC macrophages, suggesting trafficking defect that may preclude myelin turnover. Scale bars: 25 µm and 5 µm (enlargements).

addition to oligodendrocytes, hyperactive microglia may contribute to aberrant myelination in NPC.

NPC1 loss of function has been associated with impairment in lysosomal proteolysis[25,52]. However, the observed defects in degradation of myelin in *Npc1*$^{-/-}$ microglia and NPC patient macrophages are in contrast to what we observed for Aβ and EGFR where endocytosed material was degraded efficiently, supporting that lysosomal degradation function in myeloid cells may be preserved upon loss of NPC1. Along these lines, preserved lysosomal function has been anticipated in NPC patient cells[111,112]. We hypothesize that myelin degradation is impaired because phagocytosed material accumulates within late endosomes/MVBs that do not fuse with lysosomes. Impaired autophagy, vesicular fusion, and MVB accumulation has been demonstrated in NPC[22,23,111–114]. Protein substrates (e.g., Aβ or EGFR) may follow an alternative route to reach the lysosome and are therefore efficiently degraded, supporting our hypothesis of preserved lysosomal degradation in microglia. Furthermore, because the main lipid storage burden in NPC may not be occurring in lysosomes, but rather in late endosomes/MVBs, the term lysosomal storage disease may be revisited. Myelin accumulation that we observed in *Npc1*$^{-/-}$ microglia may be occurring in CD63 positive compartments, as this MVB/exosome marker is the most significantly altered protein in *Npc1*$^{-/-}$ microglia. CHM2A, that is another MVB protein, is also upregulated in pre-symptomatic phase of NPC, strengthening the relevance of MVB function for early pathogenesis of NPC[115].

Lipids deriving from myelin degradation are normally stored in lipid droplets that play a key regulatory function in cellular metabolism[77]. Beyond their important metabolic role, lipid droplets can also prevent lipo-toxicity[116,117] or protect cells from high reactive oxygen species (ROS)[118–120], ER stress or mitochondrial damage[77,121]. Thus, tuning of lipid droplet cellular content is a crucial physiological process since either disturbances in formation[119] or an excess of lipid droplets in microglia[122] may contribute to increased cellular stress and neurotoxicity. Accordingly, increased ROS, cellular stress, and mitochondrial dysfunction have been associated with NPC[123]. Moreover, aberrant lipid droplet formation[124] may also be linked to autophagy defects described in NPC[21–24,111]. It has been already shown that low concentrations of cyclodextrin treatment mobilize cholesterol from late endosomes/lysosomes and can delay neurodegeneration in vivo[125,126]. Along these lines, our MβCD rescue experiment provides a further mechanistic insight that lowering cholesterol levels partially rescues lipid droplet formation and restores

homeostasis of *Npc1*$^{-/-}$ microglia, supporting the therapeutic benefits of cholesterol lowering in NPC[11,50]. Our proteomic analysis identified autophagosome formation and maturation as the most affected pathway in microglia upon loss of NPC1. Significant proteomic changes in proteins involved in autophagy further strengthen the regulatory role of NPC1 in intracellular trafficking[22,23,111,112]. Alterations in mTOR signaling were also found in our macrophage proteome analysis, confirming that this defect occurs in NPC patient cells.

The increased lipid burden in *Npc1*$^{-/-}$ microglia may contribute to further upregulation of late endosomal/lysosomal proteins in order to compensate for the lipid degradation failure. This negative feedback loop likely results in enhanced phagocytic uptake that further amplifies lipid burden and generates a chronic and self-sustained microglial activation that may be harmful to neurons. From a therapeutic point of view, we believe that a pharmacological approach based on a combination of immune modulation and lipid reducing agents should be considered as a treatment strategy in NPC.

## Methods

**Animals**. Male and female C57BL/6J (Jackson Laboratory stock N° 000664), BALB/cNctr-Npc1$^{m1N}$/J (Jackson Laboratory stock N° 003092)[12,13], C57BL/6-Npc1tm1.2Apl[69], B6.Cx3cr1tm1.1(cre)Jung/N[70] and APPPS1 mice[127] were used in this study. Animals were group housed under specific pathogen-free conditions. Mice had access to water and standard mouse chow (Ssniff Ms-H, Ssniff Spezialdiäten GmbH, Soest, Germany) ad libitum and were kept in a 12/12-h light–dark cycle in IVC System at the temperature of 20–22 °C and humidity of 45-46%. All experimental procedures were performed in accordance with the German animal welfare law and approval for this work has been issued to DZNE by the government of Upper Bavaria (license number ROB-55.2-2532.Vet_02-17-075).

**Human material**. All studies were performed in accordance with the 1964 Declaration of Helsinki and were approved by the local Ethics Committee of the University of Munich. Seven clinically affected *NPC1* mutation carriers and three healthy volunteers (CTR) were included into this study. All participants gave written informed consent prior to their inclusion in the study. Data including *NPC1* mutation status, pharmacological regimen and clinical features of participants included into the study are summarized in Supplementary Data 5.

**Histology**. Eight weeks old animals were transcardially perfused with 4% PFA/0.1 M PBS followed by 2 h post fixation with 4% PFA/0.1 M PBS. Postnatal day 7 (P7) brains were fixed for 6 h in 4% PFA/0.1 M PBS. Free-floating 30 µm sagittal brain sections were permeabilized and blocked for 1 h in PBS/0.5% Triton X-100/5% normal goat serum (NGS). Next, samples were incubated O/N at 4 °C with primary antibody solution in blocking buffer. After washing 3 times with PBS/0.2% Triton X-100, corresponding goat secondary antibodies (Alexa Fluor, Invitrogen) were diluted 1:500 in PBS/0.2% Triton X-100/2% NGS and incubated for 1 h at RT. Nuclear staining (Hoechst 33342 1:2000, Invitrogen) or myelin dye (Fluoromyelin

red 1:300, Invitrogen) were incubated together with secondary antibodies. Sections were washed twice with PBS/0.2% Triton X-100 and once with PBS before being mounted using Fluoromount (Sigma Aldrich). Imaging was performed on 3 mice per genotype and time point using a Leica SP5 confocal microscope. Primary antibodies and dilutions were as follows: Calbindin (1:500, Swant); CD68 (1:500, AbD Serotec); Iba1 (1:300, Wako); NeuN (1:500, Millipore); CNPase (1:300, Abcam) and Synaptophysin (1:500, Abcam).

**Primary microglia**. After the brain had been dissected, cerebellum, olfactory bulb, and brain stem were removed. The remaining cerebrum was dissociated using the Neural Tissue Dissociation Kit (P) (Miltenyi Biotec) according to the manufacturer's protocol. To positively select microglial cells, cell suspension was incubated with CD11b microbeads (Miltenyi Biotec) and pulled down using a MACS separation column (Miltenyi Biotec). Purity of microglial fraction was tested by western blot using microglial, neuronal, astrocytic and oligodendrocyte markers as described below (western blot analysis). For mass spectrometry analysis, microglial fraction was washed twice with PBS and subsequently analyzed without culturing (acutely isolated). For cell culture experiments, isolated microglia were resuspended and plated at a density of $10^6$ cells/6 cm dish in microglia culturing medium containing DMEM/F12 (Gibco) supplemented with 10% fetal bovine serum (FBS, Sigma Aldrich) and 1% PenStrep (Gibco). For adult microglia, medium was supplemented with 10 ng/ml GM-CSF (R&D System). To analyze cholesterol load, microglia from WT and $Npc1^{-/-}$ animals were plated onto glass coverslips and cultured for 5 days in vitro (5DIV). Cells were fixed with 4% PFA/sucrose for 15 min at RT. After permeabilization with PBS/0.1% Triton X-100, cells were incubated for 1 h in blocking solution (2% BSA/2% FBS/0.2% fish gelatin). Afterward, primary antibody against CD68 (1:500, AbD Serotec) was incubated for 1 h at RT. After washing, cells were incubated with a goat anti-rat Alexa Fluor secondary antibody (1:500, Invitrogen) together with a cholesterol binding dye (Filipin 100 μg/ml, Sigma Aldrich)[62] for 1 h at RT. Cells were washed, mounted using Fluoromount and analyzed by confocal microscopy. Cells were analyzed from at least three independent experiments. For MβCD treatment, both WT and $Npc1^{-/-}$ P7 microglia were cultured at a density of $4 \times 10^5$ cells/well in a 12-well plate in microglia culturing medium. At 5DIV, cells were treated with 250 μM MβCD (Sigma Aldrich) or distilled water (Gibco) as a control for 48 h.

**Isolation of peripheral blood monocyte-derived macrophages**. Blood samples (20 ml) from clinically affected homozygous *NPC1* mutation carriers and healthy donors were collected. Negative selection of peripheral blood monocyte-derived macrophages was performed by incubating full blood for 20 min at RT with RosetteSep Human Monocyte Enrichment Cocktail (StemCell Technologies). An equal volume of washing buffer (D-PBS/2% FBS/1 mM EDTA) was added to each sample and layer of macrophages was separated from red blood cells and plasma by centrifugation on a Ficoll gradient ($800 \times g$ for 15 min, GE Healthcare). Potential contamination by red cells was eliminated by incubating cell pellets with ACK lysis buffer (Gibco) for 3 min at RT. Lysis buffer was quenched with 40 ml of washing buffer and cells were centrifuged at $300 \times g$ for 7 min. Cell pellets were resuspended and plated in macrophage complete medium (RPMI 1640/10% FBS/1% PenStrep/ 1X Pyruvate/1× NEAA) supplemented with 50 ng/ml hM-CSF (Thermo Scientific). After 48 h, 50 ng/ml of fresh hM-CSF was re-added. At 5DIV, media was discarded and adherent cells were washed once in PBS, incubated for 3 min at RT with Versene (Lonza) and scraped in 5 ml macrophage complete medium for further analysis.

**Sample preparation for mass spectrometry**. The microglia enriched fractions ($n = 3$) or human macrophages (7 NPC patients and 3 healthy controls) were lysed in 200 μl of STET lysis buffer (50 mM Tris/150 mM NaCl/2 mM EDTA/1% Triton X-100, pH 7.5) as described before[63,128] and incubated 15 min on ice with intermediate vortexing. The samples were centrifuged for 5 min at $16,000 \times g$ at 4 °C to remove cell debris and undissolved material. The supernatant was transferred to a fresh protein LoBind tube (Eppendorf) and the protein concentration was estimated using the Pierce 660 nm protein assay (ThermoFisher Scientific). A protein amount of 15 μg was subjected to tryptic protein digestion using the filter aided sample preparation protocol (FASP)[129] using Vivacon spin filters with a 30 kDa cut-off (Sartorius). Briefly, proteins were reduced with 20 mM dithiothreitol and free cysteine residues were alkylated with 50 mM iodoacetamide (Sigma Aldrich). After the urea washing steps, proteins were digested with 0.3 μg LysC (Promega) for 16 h at 37 °C followed by a second digestion step with 0.15 μg trypsin (Promega) for 4 h at 37 °C. The peptides were eluted into collection tubes and acidified with formic acid (Sigma Aldrich). Afterward, proteolytic peptides were desalted by stop and go extraction (STAGE) with self-packed C18 tips (Empore)[130]. After vacuum centrifugation, peptides were dissolved in 2 μl 0.1% formic acid (Biosolve) and indexed retention time peptides were added (iRT Kit, Biognosys).

**LC-MS/MS analysis**. For label-free protein quantification (LFQ), peptides were analyzed on an Easy nLC 1000 or 1200 nanoHPLC (Thermo Scientific) which was coupled online via a Nanospray Flex Ion Source (Thermo Scientific) equipped with a PRSO-V1 column oven (Sonation) to a Q-Exactive HF mass spectrometer (Thermo Scientific). An amount of 1.3 μg of peptides was separated on in-house

packed C18 columns (30 cm × 75 μm ID, ReproSil-Pur 120 C18-AQ, 1.9 μm, Dr. Maisch GmbH) using a binary gradient of water (A) and acetonitrile (B) supplemented with 0.1% formic acid (0 min, 2% B; 3:30 min, 5% B; 137:30 min, 25% B; 168:30 min, 35% B; 182:30 min, 60% B) at 50 °C column temperature.

Data-dependent acquisition (DDA) was used for LFQ of microglia from $Npc1^{-/-}$ mice and human macrophages. Full MS scans were acquired at a resolution of 120,000 ($m/z$ range: 300–1400; automatic gain control (AGC) target: 3E+6). The 15 most intense peptide ions per full MS scan were selected for peptide fragmentation (resolution: 15,000; isolation width: 1.6 $m/z$; AGC target: 1E+5; normalized collision energy (NCE): 26%). A dynamic exclusion of 120s was used for peptide fragmentation.

Data independent acquisition (DIA) was used for LFQ of microglia from $Npc1^{flox/cre}$ mice. One scan cycle included a full MS scan ($m/z$ range: 300–1400; resolution: 120,000; AGC target: 5E+6 ions) and 30 MS/MS scans covering a range of 300–1400 $m/z$ with consecutive m/z windows (resolution: 30,000; AGC target: 3E+6 ions). The maximum ion trapping time was set to "auto". A stepped NCE of 26% ± 2.6% was used for fragmentation.

**Mass spectrometry data analysis and LFQ**. Proteome from $Npc1^{-/-}$ microglia was analyzed with the software Maxquant, version 1.6.3.3 (maxquant.org, Max-Planck Institute Munich)[131]. The MS data were searched against a reviewed canonical fasta database of *Mus musculus* from UniProt (download: November the 5th 2018, 17,005 entries). Trypsin was defined as a protease. Two missed cleavages were allowed for the database search. The option first search was used to recalibrate the peptide masses within a window of 20 ppm. For the main search peptide and peptide fragment mass tolerances were set to 4.5 and 20 ppm, respectively. Carbamidomethylation of cysteine was defined as static modification. Acetylation of the protein N-terminal as well as oxidation of methionine were set as variable modifications. The false discovery rate for both peptides and proteins was adjusted to less than 1%. The "match between runs" option was enabled with a matching window of 1.5 min. LFQ of proteins required at least one ratio count of unique peptides. Only unique peptides were used for quantification. Normalization of LFQ intensities was performed separately for the age groups because LC-MS/MS data were acquired in different batches. The protein LFQ reports of Maxquant were further processed in Perseus[132]. The protein LFQ intensities were log2 transformed and log2 fold changes were calculated between NPC-deficient and WT samples separately for the different age groups, mouse models and patients. LFQ intensities were detected in every biological replicate in order for the proteins to be considered quantifiable and no imputation was performed to replace missing values. An unpaired Student's $t$-test with two-tailed distribution was applied to evaluate the significance of proteins with changed abundance. Additionally, a permutation-based false discovery rate (FDR) estimation (threshold: FDR = 5%) was used to perform multiple hypothesis correction[133]. To ensure the adequate FDR cutoff, we experimentally determined the s0 values separately for each experimental dataset. Briefly, the "relative difference" $d_i$ of each protein is calculated, which includes s0 as a variable. Next, s0 was optimized separately for each experimental dataset, based on an interpolation to reach the lowest coefficient of variation for the absolute $d_i$ values[133]. A log2 fold change larger than 0.5, or smaller than −0.5, a $p$-value less than 0.05 and significant regulation after FDR filtering were defined as regulation thresholds criteria.

The proteomic data was further analyzed through the use of ingenuity pathway analysis (IPA, QIAGEN Inc., https://www.qiagenbioinformatics.com/products/ ingenuity-pathway-analysis). Standard settings were used for the analysis. Proteins with a log2 fold-change larger than 0.5, or smaller than −0.5 and a $p$-value less than 0.05 were defined as protein regulation thresholds.

Proteome from human macrophages was analyzed with Maxquant version 1.6.3.3 using the same settings searching against a reviewed fasta database of *Homo sapiens* from UniProt including isoforms (download: December the 17th 2018, 42,432 entries).

Proteome from $Npc1^{flox/cre}$ microglia was analyzed with the software Spectronaut (version 12.0.20491.14.21367) with standard settings using our previously generated microglia spectral library, including 122,542 precursor ions from 91,349 peptides, which represent 6223 protein groups[63]. Briefly, the FDR of protein and peptide identifications was set to 1%. LFQ of proteins was performed on peptide fragment ions and required at least one quantified peptide per protein. Protein quantification was performed on at least 2 and maximum 3 peptides per protein group. A permutation-based FDR correction for multiple hypotheses was applied as described above.

**Western blot analysis**. Acutely isolated and cultured microglia or human macrophages were lysed in STET lysis buffer supplemented with protease and phosphatase inhibitors (Sigma Aldrich). Lysate protein content was quantified using Bradford assay (Biorad) according to the manufacturer´s protocol. At least 10 μg per sample were separated on a bis-tris acrylamide gel followed by western blotting either on a PVDF or nitrocellulose membrane (Millipore) using the following antibodies: Iba1 (1:1000, Wako); GFAP (1:1000, Dako); Tuj1 (1:1000, Covance); CNPase (1:1000, Abcam); NPC1 (1:1000, Abcam); NPC2 (1:1000, Sigma Aldrich); LAMP1 (1:1000, Sigma Aldrich); Cathepsin B (1:2000, R&D System); Cathepsin D (1:1000, Novus Biologicals); mouse CD68 (1:1000, AbD Serotec); human CD68 (1:500, Acris); mouse CD63 (1:1000, Abcam); mouse GRN (1:50, clone 8H10,[85]);

human GRN (1:1000, Invitrogen); TREM2 (1:10, clone 5F4[66]); APOE (1:1000, Millipore); PLP1 (1:1000, Abcam); EGFR (1:1000, Abcam) and TGFB1 (1:1000, R&D System). Blots were developed using horseradish peroxidase-conjugated secondary antibodies (Promega) and the ECL chemiluminescence system (GE Healthcare). An antibody against Calnexin (1:1000, Stressgen) or GAPDH (1:2000, Abcam) was used as loading control. Blot densitometry quantification was performed using gel analyzer tool in ImageJ (NIH) with at least $n = 3$ for both human and murine samples.

**Ex vivo Aβ plaque clearance and myelin phagocytic assay**. To functionally characterize myeloid cells (primary microglia and human macrophages), we adapted a previously reported phagocytic assay[64]. Briefly, a 10 μm brain section from an AD mouse model (APPPS1) was placed on a poly L-lysine coated glass coverslip. In order to stimulate cell recruitment to the plaque site, brain sections were incubated for 1 h at RT with an antibody against human Aβ (5 μg/ml 6E10, BioLegend for microglia and 3 μg/ml 2D8[134] for macrophages). Isolated cells were plated at a density of $3 \times 10^5$ (microglia) or $2.5 \times 10^5$ (macrophages) cells/coverslip and incubated either for 5DIV (microglia) or 1DIV (macrophages) in corresponding culturing medium. Next, samples were fixed with a 4% PFA/sucrose solution for 15 min at RT. Immunostainings were performed as described above using an antibody against myeloid cells (anti-CD68 for microglia and anti-CD45 (1:300, Abcam) for macrophages) and Aβ plaque (1:500, 3552,[63]). Fibrillar Aβ (plaque core) was visualized using Thiazine Red (ThR, 2 μM, Sigma Aldrich) added into the secondary antibody solution. To evaluate the phagocytic capacity of myeloid cells, plaque coverage (ThR signal area for microglia or 3552 for macrophages) was quantified, comparing brain section incubated with cells with a consecutive brain section where no cells were added. Quantification was done using a macro tool in ImageJ (NIH), applying to 10×16-bit tail scan picture of the whole coverslip with a threshold algorithm (OTSU) and measuring particles with a pixel size from 5 to infinity. For microglia, each experimental group was tested in three independent experiments, while for macrophages 6 NPC patients and 3 healthy controls were analyzed.

Similarly, we applied this assay to evaluate the capability of P7 microglia to uptake and digest myelin from the APPPS1 brain section. In order to evaluate myelin level after incubation with microglia, we used Fluoromyelin dye (1:300, Invitrogen) added together with the secondary antibody solution. CD68 was used to visualize microglia. Each experimental group was tested in three independent experiments.

**In vitro myelin phagocytic assay**. Myelin was isolated from 8-weeks old C57BL/6J mouse brains which were homogenized by sonication in 10 mM HEPES/5 mM EDTA/0.3 M sucrose/protease inhibitors[76]. The homogenate was layered on a gradient of 0.32 M and 0.85 M sucrose in 10 mM HEPES/5 mM EDTA (pH 7.4) and centrifuged at $75,000 \times g$ for 30 min with a SW41 Ti rotor (Beckman Coulter). The myelin fraction was isolated from the interface, and subjected to three rounds of osmotic shock in sterile ultrapure water and centrifuged at $75,000 \times g$ for 15 min. The resulting pellet was subjected to the same procedure to obtain a pure myelin fraction. The yield of myelin was calculated by measuring the total amount of protein with the Bradford assay (Biorad). For the fluorescence labeling of myelin, a PKH67 Green Fluorescent Cell Linker Mini Kit was used (Sigma Aldrich).

Primary microglia from P7 mice and human macrophages were isolated and cultured as described above. Cells were fed with purified green fluorescently labeled myelin at a concentration of 50 μg/ml. At 6 h, myelin was washed out and cells were fixed at 24/48/72 h with a 4% PFA/sucrose solution for 15 min. Immunostaining of fixed cells was performed as described above using antibodies against CD68 and Perilipin 2 (1:200, Progen) for microglia and CD45 for patient macrophages. Microglial lipid droplets were visualized using Nile red staining kit (Abcam) according to the manufacturer's protocol. In order to monitor myelin trafficking during phagocytosis, primary P7 microglia were incubated with cholera toxin (CTX) subunit B Alexa Fluor™ 647 Conjugate (0.5 μg/ml, Thermo Fisher Scientific) and Red DQ-bovine serum albumin (DQ-BSA; 10 μg/ml, Invitrogen) for 30 min at 37 °C in microglial culturing medium. Cells were pulsed for 15 min with green fluorescently labeled myelin (10 μg/ml) and fixed after 1 h. For microglia rescue experiments, after myelin wash out, microglia were treated with 250 μM MβCD (Sigma Aldrich) or with distilled water (Gibco) as a control for 48 h and fixed with 4% PFA/sucrose for 15 min. For the quantification of the rescue experiment, 300 cells per technical replicate were analyzed (three technical replicates/cell preparation). Cells with Nile red positive structures in cell periphery were defined as lipid droplet positive cells. MβCD cholesterol-reducing activity was tested using a cholesterol-binding dye (Filipin 100 μg/ml, Sigma Aldrich). For all microglia experiments, three independent cell preparations ($n = 3$) were treated and analyzed by confocal microscopy. For human macrophages, cells from 2 NPC patients and 2 healthy controls were analyzed.

**EM analysis**. The microglial pellet was preserved throughout all fixation, contrasting and infiltration steps. Cells were fixed for 30 min in 2% PFA/2.5% glutaraldehyde (EM-grade, Science Services) in 0.1 M sodium cacodylate buffer (pH 7.4) (Sigma Aldrich), washed three times in 0.1 M sodium cacodylate buffer before

postfixation in reduced osmium (1% osmium tetroxide (Science Services)/0.8% potassium ferrocyanide (Sigma Aldrich) in 0.1 M sodium cacodylate buffer. After contrasting in aqueous 0.5% uranylacetate (Science Services), the pellet was dehydrated in an ascending ethanol series, infiltrated in epon (Serva) and cured for 48 h at 60 °C. Ultrathin sections (50 nm) were deposited onto formvar-coated copper grids (Plano) and postcontrasted using 1% uranyl acetate in water and ultrostain (Leica). Transmission Electron Microscopy micrographs were acquired on a JEM 1400plus (JEOL) using the TEMCenter and tile scans with the Shot-Meister software packages (JEOL), respectively. Three independent cell preparations were analyzed ($n = 3$).

**EGFR degradation assay**. Both WT and $Npc1^{-/-}$ P7 microglia were cultured at a density of $5 \times 10^5$ cells/well in a 12-well plate in microglia culturing medium. At 5DIV, microglia culturing medium was replaced with serum-free medium, containing only DMEM/F12 and 1% PenStrep. After 3 h of serum deprivation ($t = 0$), cells were treated with mouse recombinant EGF (40 ng/ml, R&D System) and cycloheximide (20 μg/ml, Sigma Aldrich) in serum-free medium. Cells were lysed at different time points (0, 1, 3, and 6 h) after treatment in STET lysis buffer and supplemented with protease and phosphatase inhibitors. EGF receptor (EGFR) level in each lysate was analyzed via western blot using an antibody against EGFR (anti-EGFR antibody EP38Y, Abcam). Each experimental group was tested in microglia isolated from three independent experiments ($n = 3$).

**Statistical analysis**. For comparison between two groups, unpaired Student's $t$-test with two-tailed distribution was used. For the in vitro phagocytic assay with MβCD, one-way ANOVA followed by unpaired two-tailed Student's $t$-test was used. Data are represented as mean ± standard error of the mean (SEM). A value of $p < 0.05$ was considered significant ($*p < 0.05$; $**p < 0.01$; $***p < 0.001$).

**Reporting summary**. Further information on experimental design is available in the Nature Research Reporting Summary linked to this paper.

## Data availability

The mass spectrometry proteomics data are deposited at the ProteomeXchange Consortium via the PRIDE partner repository with the dataset identifier PXD019447 (proteomic signature of $Npc1^{-/-}$ microglia), PXD019452 (proteomic signature of $Npc1^{flox/Cre}$ microglia) and PXD020659 (proteomic signature of NPC human macrophages). Source data are provided with this paper.

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

## Acknowledgements

We thank Anita Hennig and Baccara Hizli for help with human blood sample collection and Anna Berghofer for excellent technical assistance. APPPS1 mice were kindly provided by Mathias Jucker (Hertie-Institute for Clinical Brain Research, University of Tübingen) and B6.Cx3cr1tm1.1(cre)Jung/N mice by Steffen Jung (Weizmann Institute of Science). The authors thank Anja Capell, Dieter Edbauer, Christian Haass, Matthias Prestel, and Michael Willem for critically reading the manuscript. This work was supported by the NCL Foundation, the Alzheimer Forschung Initiative e.V. (grant number 18014), an Alzheimer's Association Grant through the AD Strategic Fund (ADSF-21-831226-C), the BMBF through JPND PMG-AD and the Deutsche Forschungsgemeinschaft (DFG, German Research Foundation) within the framework of the Munich Cluster for Systems Neurology (EXC 2145 SyNergy; ID 390857198). A.P.L. was supported by the N.I.H. (R01 NS063967). S.H. was supported by the Croatian Science Foundation (project IP-2016-06-2799). S.A.S. was supported by a LMU Excellence Program for Clinician Scientists, Verum-Stiftung and the Ara Parseghian Medical Research Foundation.

## Author contributions

S.T. and A.C. designed and supervised the study and wrote the manuscript with the input from all co-authors. A.C. and L.D. performed human and animal experiments, including histology, microglia and human macrophage experiments, target validation and functional studies. A.P.L. provided the *Npc1<sup>flox</sup>* mouse model. S.H. contributed to conceptual design. L.V. contributed to histological data. L.S.M. assisted in isolation of primary microglia. M.Sch. performed EM analysis of isolated microglia. L.C.C and M.S. contributed to the design of the in vitro myelin phagocytic assay. S.A.M., J.K., and S.F.L. performed the proteomic analysis. T.B.E., S.A.S. and M.Str. recruited NPC patients and control individuals and delivered study samples.

## Competing interests

M.Str. is Joint Chief Editor of the Journal of Neurology, Editor in Chief of Frontiers of Neuro-otology and Section Editor of F1000. He has received speaker's honoraria from Abbott, Actelion, Auris Medical, Biogen, Eisai, Grünenthal, GSK, Henning Pharma, Interacoustics, Merck, MSD, Otometrics, Pierre-Fabre, TEVA, UCB. He is a shareholder of IntraBio. He acts as a consultant for Abbott, Actelion, AurisMedical, Heel, IntraBio, and Sensorion. T.B.E. served as a consultant for Actelion and Sanofi-Genzyme. All other authors declare that they have no competing interests.
