## [Peer Review File · Nature Communications]

Reviewers' comments:

Reviewer #1 (Remarks to the Author):

The manuscript by Colombo et al. reports phenotype characterization of microglia from presymptomatic and symptomatic NPC1-deficient mice, providing further evidence into the causal role of microglia (prior to neuronal loss and disease-related behavioral outcomes) observed in this murine model of Niemann-Pick type C (NPC) disease. Specifically, the authors utilize mass spectrometry-based proteomics and several orthogonal approaches (e.g., imaging and phagocytosis assay) to show that the microglial activation phenotype in presymptomatic mice (P7 in NPC mouse model) is characterized by increased phagocytosis as well as differential expression of proteins associated with autophagy and phagosome-related pathways. The study shows that increased microglial phagocytosis affects myelin uptake (but with impaired processing) and also implicates this phenotype to altered synaptic pruning during early stages of the disease model. Several points of this manuscript are corroborated by previous studies including a recent publication (Hum Mol Genet, 27(12), 2018, 2076–2089) that shows increased microglial phagocytosis and possible evidence of phagocytosed neurons (CD22) in NPC1-deficient mice – although in later stages of this disease model. Moreover, the authors in this study also suggest peripheral macrophages from NPC patients as a potential source for disease biomarker development. The results from this study provide valuable molecular detail and potential mechanisms for early-stage microglial involvement in NPC disease development; however, several issues need to be addressed as outlined below:

1) The authors report a label-free quantitative proteomic approach to compare WT to NPC1-deficient mice at P7 and 8 weeks of age. Clarification is needed for the data analysis/statistical filtering. For example, does the statement “only proteins with a consistent quantification for all three samples per age group” mean that LFQ intensities have to be detected in every biological replicate in order for the proteins to be considered quantifiable (i.e., no imputation was performed to replace missing values)? In regard to the statistical filtering, the authors use the default s_0 value (0.1) in Perseus when generating a volcano plot. A non-zero s_0 value applies a secondary filtering step that considers t-test diff/fold change, along with p-value from t-test, to control FDR. This parameter is likely experiment dependent and there is no description of why this value was specified. After inspection of the proteome dataset (for 8 weeks comparison), statistical significance with FDR is still achieved at ~20% expression change. It is surprising that this effect size is possible to detect with adequate statistical power at only three biological replicates. The authors need to provide details of the quantitation precision of their experiments for proteomic analysis of acutely isolated microglia using the FASP and label-free workflow. This would provide rationale for their reported statistical filtering. The proteins selected for western validation represent those with higher fold change values and $-\log(p\text{-values})$, which would presumably be in a region with lower expected FDR anyway. The seemingly arbitrary \log_2 fold-change cutoff of ± 0.5 used as a filter for IPA analysis could actually have relevance in terms of controlling FDR (along with p-value); however, there was no statistical rationale for this fold change selection. Also, related to IPA analysis, it was not clear if FDR-corrected p-values were utilized (e.g., Benjamini-Hochberg p-value in IPA). If not, no rationale was provided.

The authors should provide the complete list of proteins along with intensity values for each protein (e.g., the ProteinGroups file from MaxQuant) and also submit the raw data to a public repository (e.g., ProteomeXchange). Although the authors provide assessment of purity of isolation by western analysis of cell type-specific markers, access to the complete list of proteins from proteomic analysis provides the reader another means to assess possible contamination as well as other areas of interpretation.

In terms of validation of the proteomics results, the authors demonstrate western analysis for certain proteins related to microglial phenotype. Antibody vendor information was provided; however, no information on how the authors authenticated the specificity of the antibodies was described. This issue is more of a concern for the reported westerns that show multiple bands. Also, calnexin was used as a loading control – are the data shown a representative blot that was

consistently obtained for every gel (if so, at least provide mean quantitative value and associated variance) or were the westerns run as multi-plex analysis (if so, how was cross-reactivity minimized for so many proteins probed)? There was no quantitation and the number of replicates was low (n=1-2). Qualitative assessment of band patterns does not adequately address scientific rigor in the experimental design (when using western blot as independent method of validation).

2) The results shown for bioinformatic analysis by IPA was a weakness. The authors only report enriched canonical pathways from IPA. There are additional meaningful results that can be obtained including the prediction of upstream regulator activity along with significance of overlap to the proteome dataset. Additionally, biological/disease functional outcomes can be predicted and degree of enrichment determined. In terms of functional outcomes relevant to the reported phenotype, significant enrichment and activation of phagocytosis would be observed even for P7 (e.g., EHD1, CORO1C, SCARB2, LGALS3, LAMP1, LAMP2, GSN, GRN). The proteomic data show highly significant overlap with known downstream targets of TGFB1 and TFEB. Interestingly, the proteomic data show downregulation of TGFB1 at 8 weeks of age – perhaps indicating one possibility for inflammatory phenotype observed in symptomatic time period. Additionally, IPA includes an analysis match function that quantitatively compares the submitted proteomic data with published datasets. This could be useful to support reported DAM phenotype to other relevant phenotypes observed in various diseases/disorders of the CNS.

3) Although the use of peripheral macrophage proteome profiles as a biomarker for NPC is novel, the sample size in this study (n=2 NPC patients) is extremely limited to make any solid conclusions. Based on the proteomic data and western analysis shown, there appears to be overlap and more subtle changes (compared to the control patient) associated with the panel of markers selected. It is understandable that access to NPC patient samples is limited (although the recent paper in Hum Mol Genet had a decent sample size when assessing CD22 as potential marker), but assuming a correlation with age/disease progression with two patient samples is over-interpreting the data. The authors would have to do more work to determine specificity and sensitivity of the biomarker panel.

4) The authors show imaging and proteomic data indicating phagocytosed neuronal material within microglia during presymptomatic stages of the NPC1-deficient mouse model. The authors suggest altered synaptic pruning from increased microglial phagocytosis. Are there structural features to further support this – for example, changes in spine density or other altered neuronal structural features that can be attributed to microglia (and also distinguish from dysfunction arising from lipid trafficking defect in NPC1-deficient neurons)? Also, given the role of astrocytes in signaling microglia for synapse remodeling [e.g., through IL-33: Science, 6381, 1269-1273 (2018)], how does this integrate into the interpretation of the ex vivo results?

5) The authors show an interesting microglial phenotype that could explain a causal role of microglia in early stages of NPC disease. Interestingly, several markers identified from the proteomic analysis (P7) show overlap with downstream targets affected by HPβCD treatment (CD68, NPC2, CTSB, CTSD). Additionally, a previous study shows that HPβCD treatment in an earlier time period (presymptomatic stage) in NPC1-deficient mice is more effective in terms of reducing microglial activation and improving lifespan than later stage administration (Hum Mol Genet, 27(12), 2018, 2076–2089). Given the authors have potential protein markers associated with DAM phenotype in early stages of the disease model, the efficacy of HPβCD treatment or other pharmacological approach related to rescuing phenotype based on proteomic data would provide more value to this study.

Reviewer #2 (Remarks to the Author):

This manuscript by Columbo et al describes the proteomic characterization of microglia from Npc1 mutant mice. They are able to show a protein expression pattern consistent with the Disease Associated Microglia signature pattern. This change occurs early in the disease process. They functionally characterize microglia from Npc1 mutant mice and demonstrate increased uptake but impaired turnover of myelin. They also show enhance processing of AB. The paper then

demonstrates similarities between mouse microglia and patient peripheral macrophage. Based on these similarities they conclude that peripheral macrophage can be used to monitor disease progression and therapeutic interventions

This paper is well written and does contribute to our understanding of the potential role of microglia in NPC1 pathology. This is the first report of microglial proteomics in NPC1 but the actual findings are primarily supportive of prior work. The human work is extremely weak and the idea of utilizing peripheral macrophage to monitor disease progression and therapeutic interventions simply is not well supported.

1. Human gene nomenclature should be used when referring to the human gene. NPC1 and NPC2 not Npc1 and Npc2. The latter being mouse nomenclature.
2. Colocalization of filipin with lamp1 would be more typical than using CD68 to demonstrate late endosome/lysosomal localization.
3. Line 126. Previous work (Peake 2011 reference) has shown that Npc1 mutant microglia still display a cholesterol storage phenotype.
4. Line 178. Authors should address overlap of their findings with those previously described by Alam et al JBC 2014 289:8051-8066.
5. Line 211. Authors should address prior work describing increased myelin uptake Gabande-Rodriguez EMBO J 2019 E99553.
6. 1/100,000 is a better estimate of live birth rate.
7. Not clear how one obtains a "good correlation" when human samples are limited to an n of 2. Figure 7A only shows a change with age yet the text talks about pathological stage (what exactly are they referring to) and clinical symptoms (which ones). Describing NPC1 disease progression as typical is not useful. What are the potential effects of the concomitant medications on the proteomic study? Why was the validation only done with one of the two patient samples?
8. The idea that synaptic pruning is impaired needs further support.
9. Line 299. It is not "striking" to find similarities between macrophage and microglia
10. Only one healthy control is used and that control is significantly older than either of the patients.
11. Lack of perfusion could lead to contamination of the microglial preparation with peripheral mononuclear cells.
12. Is a student t-test appropriate for experiments with n=3? Would prefer authors to show SD rather than SEM. Would prefer authors show actual data points with the bar graphs. Provide information as to what statistical program was used.
13. Abstract. The consequences of loss of NPC1 function on microglia may not be fully described, but it is not unknown.
14. The authors should discuss why the clearance rates of AB and myelin are different.

Work that should be addressed in more detail.

<https://www.ncbi.nlm.nih.gov/pmc/articles/PMC6331723/>
<https://www.ncbi.nlm.nih.gov/pmc/articles/PMC3308731/>
<https://www.ncbi.nlm.nih.gov/pmc/articles/PMC5985727/>
<https://www.ncbi.nlm.nih.gov/pmc/articles/PMC5532464/>
<https://www.ncbi.nlm.nih.gov/pmc/articles/PMC3961638/>

Reviewer #3 (Remarks to the Author):

Niemann-Pick type C disease is a rare neurodegenerative disorder mainly caused by mutations in Npc1, resulting in abnormal late endosomal/lysosomal lipid storage. Here, the authors provide microglial proteomic signatures and phenotypes in a NPC1-deficient (Npc1^{-/-}) murine model and patient blood-derived macrophages. They demonstrate enhanced phagocytic uptake and impaired lipid trafficking in Npc1^{-/-} microglia that precede neuronal death. Conclusions are that NPC1 plays a key role in immune cells. This is quite an interesting study, but highly descriptive with very little mechanistic insights. Moreover, since this is a general model of Npc1^{-/-} gene deletion, how the

authors can be sure that protein changes in microglia are directly link to Npc1 in microglia and not other cell types (Oligodendrocytes)?

The data are also a little bit all over the place from Abeta and myelin phagocytosis to synaptic pruning. The link between these two events are not clear, since the clearance of toxic proteins would normally be beneficial for neurons and pruning!

In addition, the proteomic data are largely descriptive and do not provide direct mechanistic evident that these changes are related to the phagocytosis and pruning by microglia and human macrophages. Obviously, macrophages would not be involved in synaptic pruning!

Point-by-point response to reviewers' comments

Please, find below our point-by-point response that summarizes novel data and changes that are included into the revised manuscript. We hope that you appreciate that the quality and the novelty of our manuscript improved substantially to be re-considered for publication in Nature Communications.

Rewiever #1:

1. The authors report a label-free quantitative proteomic approach to compare WT to NPC1-deficient mice at P7 and 8 weeks of age. Clarification is needed for the data analysis/statistical filtering. For example, does the statement "only proteins with a consistent quantification for all three samples per age group" mean that LFQ intensities have to be detected in every biological replicate in order for the proteins to be considered quantifiable (i.e., no imputation was performed to replace missing values)? In regard to the statistical filtering, the authors use the default s0 value (0.1) in Perseus when generating a volcano plot. A non-zero s0 value applies a secondary filtering step that considers t-test diff/fold change, along with p-value from t-test, to control FDR. This parameter is likely experiment dependent and there is no description of why this value was specified. After inspection of the proteome dataset (for 8 weeks comparison), statistical significance with FDR is still achieved at ~20% expression change. It is surprising that this effect size is possible to detect with adequate statistical power at only three biological replicates. The authors need to provide details of the quantitation precision of their experiments for proteomic analysis of acutely isolated microglia using the FASP and label-free workflow. This would provide rationale for their reported statistical filtering. The proteins selected for western validation represent those with higher fold change values and -log(p-values), which would presumably be in a region with lower expected FDR anyway. The seemingly arbitrary log2 fold-change cutoff of +/-0.5 used as a filter for IPA analysis could actually have relevance in terms of controlling FDR (along with p-value); however, there was no statistical rationale for this fold change selection. Also, related to IPA analysis, it was not clear if FDR-corrected p-values were utilized (e.g., Benjamini-Hochberg p-value in IPA). If not, no rationale was provided.

As requested by the reviewer, clarification for the data analysis/statistical filtering is included in the Methods section of the revised manuscript (lines 601-658). Additionally, our detailed method development for proteomic analysis of microglia has been reported recently ¹. As anticipated by the reviewer, the statement "only proteins with a consistent quantification for all three samples per age group" means that LFQ intensities were detected in every biological replicate in order for the proteins to be considered quantifiable and no imputation was performed to replace missing values (lines 638-639). Moreover, as pointed out by the reviewer, *Npc1*^{-/-} microglia show very strong proteomic alterations that were already detectable with adequate statistical power using only three biological replicates. Nevertheless, we also took into consideration our small sample size and applied the rather conservative FDR correction of 5%, with s0 value of 0.1 (lines 641-643) that reduces false positive findings and controls for the possible inflation of the p-value ². To illustrate the high reproducibility of protein LFQ in our study, we provide an example of scatter plots of log2 transformed LFQ intensities of microglia from 3 different WT and *Npc1*^{-/-} mice at 8 weeks of age (Figure for reviewer's assessment 1). This data support the high correlation among different biological samples within the same experimental group, and demonstrate pronounced differences between WT and *Npc1*^{-/-} microglia. As pointed out by the reviewer, the log2 fold-change cutoff of +/- 0.5 was used as a filter for IPA analysis. We have defined this threshold (\cong 1.41-fold on linear scale) to avoid minor, but statistically significant changes that may be biologically less relevant. Although filtering of the proteomic data according to Benjamini-Hochberg method represents a valuable alternative, we preferred not to use this correction method since it has been suggested that, because of its highly stringent criteria, it may miss some biologically significant true

hits and increase the number of false negative proteins³. Moreover, as outlined above, while the Benjamini-Hochberg correction method relies only on p-values, we decided to also consider protein fold changes.

The authors should provide the complete list of proteins along with intensity values for each protein (e.g., the ProteinGroups file from MaxQuant) and also submit the raw data to a public repository (e.g., ProteomeXchange). Although the authors provide assessment of purity of isolation by western analysis of cell type-specific markers, access to the complete list of proteins from proteomic analysis provides the reader another means to assess possible contamination as well as other areas of interpretation.

Complete list of proteins, including averages of protein LFQ intensity ratios, log₂ conversion and corresponding p-values are provided in Supplementary Tables 1-4. The raw data are deposited to the ProteomeXchange Consortium via the PRIDE partner repository with the dataset identifier PXD019447 (proteomic signature of *Npc1*^{-/-} microglia, username: reviewer92568@ebi.ac.uk; password: BdAuYpc4), PXD019452 (proteomic signature of *NPC1*^{flox/Cre} microglia, username: reviewer06478@ebi.ac.uk; password: Yk26OI2m) and PXD020659 (proteomic signature of NPC human macrophages, username: reviewer19070@ebi.ac.uk; password: eYb15mtk). Furthermore, additional quality controls for purity of microglial isolation are included in our previous manuscript that focused on the establishment of the method¹ as additionally outlined in our response to reviewer #2 (point 11).

In terms of validation of the proteomics results, the authors demonstrate western analysis for certain proteins related to microglial phenotype. Antibody vendor information was provided; however, no information on how the authors authenticated the specificity of the antibodies was described. This issue is more of a concern for the reported westerns that show multiple bands. Also, calnexin was used as a loading control – are the data shown a representative blot that was consistently obtained for every gel (if so, at least provide mean quantitative value and associated variance) or were the westerns run as multi-plex analysis (if so, how was cross-reactivity minimized for so many proteins probed)? There was no quantitation and the number of replicates was low (n=1-2). Qualitative assessment of band patterns does not adequately address scientific rigor in the experimental design (when using western blot as independent method of validation).

Antibodies used for validation of the proteomic results have been previously reported as outlined: NPC1⁴, NPC2⁵, LAMP1⁶, CD68⁷, CD63⁸, GRN⁹, ApoE¹, PLP1¹⁰, TGFB1¹¹. In particular, antibodies showing multiple bands such as TREM2¹² or CTSB/CTSD^{9,13} are commonly used in our laboratory and have been previously validated. In case of multiple bands, band identity was specified in figure legends. As the total yield of microglial proteins obtained from one animal is rather low (about 50 µg), our membranes were re-probed. To reduce cross-reactivity, membranes were re-probed with antibodies raised in different species and when this was not possible an additional stripping step (Restore™ PLUS Western Blot Stripping Buffer, Thermo Scientific) was included. To increase the number of validated proteins per animal, loading control was run for each set of lysates, and not for each membrane. Following the reviewer's request, we have improved the western blot validation of proteomic data in the revised manuscript by increasing the number of independent animals tested (n=3) and providing a corresponding densitometry quantification of immunoblots.

2. The results shown for bioinformatic analysis by IPA was a weakness. The authors only report enriched canonical pathways from IPA. There are additional meaningful results that can be obtained including the prediction of upstream regulator activity along with significance of overlap to the proteome dataset. Additionally, biological/disease functional outcomes can be predicted and degree of enrichment determined. In terms of functional outcomes relevant to the reported phenotype,

significant enrichment and activation of phagocytosis would be observed even for P7 (e.g., EHD1, CORO1C, SCARB2, LGALS3, LAMP1, LAMP2, GSN, GRN). The proteomic data show highly significant overlap with known downstream targets of TGFB1 and TFEB. Interestingly, the proteomic data show downregulation of TGFB1 at 8 weeks of age – perhaps indicating one possibility for inflammatory phenotype observed in symptomatic time period. Additionally, IPA includes an analysis match function that quantitatively compares the submitted proteomic data with published datasets. This could be useful to support reported DAM phenotype to other relevant phenotypes observed in various diseases/disorders of the CNS.

We are thankful to the reviewer for this criticism. We performed the upstream regulator analysis of our proteomic data and, as already anticipated by the reviewer, find a master regulator of microglial homeostasis TGFB1 as significantly downregulated pathway while TFEB signaling -that is critical for autophagy/lysosome function- was upregulated in *Npc1*^{-/-} microglia at P7 (Supplementary Fig. 2d). This result is well in line with reduced homeostatic and increased DAM signatures and corresponding early microglial functional defects observed in pre-symptomatic *Npc1*^{-/-} mice. Further analysis of proteins regulated by TGFB1 is strengthening alterations of endosomal/lysosomal signaling and reduced microglial homeostasis in *Npc1*^{-/-} microglia at both pathological stages (Supplementary Fig. 2e). Our additional new mechanistic analysis reveals that lowering cholesterol in *Npc1*^{-/-} microglia results in increasing levels of TGFB1 and reducing DAM signature protein CTSD (outlined in more detail below, in our response to point 5). This effect is correlating with increased formation of lipid droplets, supporting that enhancing homeostasis of *Npc1*^{-/-} microglia may improve endosomal/lysosomal trafficking dysfunction. This data are presented in new Figure 7. Furthermore, in the revised manuscript, we compared the published transcriptomic dataset¹⁴ and here generated proteomic profiles of *Npc1*^{-/-} microglia and found a substantial overlap (Supplementary Fig. 2b), suggesting that DAM and homeostatic signatures are largely regulated at the transcriptional level, as we also observed in microglia of Alzheimer's mouse models¹.

3. Although the use of peripheral macrophage proteome profiles as a biomarker for NPC is novel, the sample size in this study (n=2 NPC patients) is extremely limited to make any solid conclusions. Based on the proteomic data and western analysis shown, there appears to be overlap and more subtle changes (compared to the control patient) associated with the panel of markers selected. It is understandable that access to NPC patient samples is limited (although the recent paper in Hum Mol Genet had a decent sample size when assessing CD22 as potential marker), but assuming a correlation with age/disease progression with two patient samples is over-interpreting the data. The authors would have to do more work to determine specificity and sensitivity of the biomarker panel.

We agree with the reviewer that our initial sample size for the proteomic analysis of patient cells was small (n=2). Accordingly, we increased the number of patient samples analyzed by proteomics to n=7 in the revised manuscript. Our proteomic changes in human macrophages are indeed still more subtle compared to rodent microglia. Nevertheless, altered endosomal/lysosomal, lipid and inflammatory signatures, including increased levels of LIPA, NPC2, GRN, ITGAX, CTSD, LAMP1/2 and APOE were detected in human macrophages that also, albeit to a lower degree, recapitulate functional changes observed in *Npc1*^{-/-} microglia. More subtle molecular and functional changes in human macrophages may be contributed by the full loss of NPC1 in a mouse model compared to its partial reduction in human patients, patient-to-patient variability or intrinsic differences between microglia and macrophages. This is discussed along lines 415-417 of our manuscript. We acknowledge the reviewer's criticism that our data are not powerful enough to make a correlation between proteomic changes and patient age/disease stage and have removed this assumption from the manuscript.

4. The authors show imaging and proteomic data indicating phagocytosed neuronal material within microglia during presymptomatic stages of the NPC1-deficient mouse model. The authors suggest altered synaptic pruning from increased microglial phagocytosis. Are there structural features to further support this – for example, changes in spine density or other altered neuronal structural features that can be attributed to microglia (and also distinguish from dysfunction arising from lipid trafficking defect in NPC1-deficient neurons)? Also, given the role of astrocytes in signaling microglia for synapse remodeling [e.g., through IL-33: *Science*, 6381, 1269-1273 (2018)], how does this integrate into the interpretation of the ex vivo results?

As the reviewer has pointed out, our data do not provide direct evidence that increased uptake of synaptic material observed in *Npc1*^{-/-} and now also in *Npc1*^{fllox/cre+} microglia is triggering neuronal structural changes. However, along these lines, microglial uptake of dendritic material has been previously reported to occur prior to obvious neuronal loss¹⁵ and reduced spine density has been detected in NPC-deficient mice *in vivo*¹⁶. Furthermore, current literature provides multiple evidence linking enhanced phagocytic uptake of microglia with alterations in synaptic pruning in other neurodegenerative conditions^{17,18}, and upon enhanced lysosomal activity¹⁹ which is in accordance with our observations. This supportive evidence is discussed along lines 432-443 of the revised manuscript. Additionally, we fully agree with the reviewer that astrocytes also play an important role during synaptic pruning and that their role in synaptic pruning in NPC should also be considered. However, as this aspect remained beyond the scope of our revised manuscript, we toned down our initial statement that NPC1 regulates synaptic pruning and, as outlined above, only discussed this possibility based on enhanced phagocytic uptake of neuronal material that we observe in NPC1-deficient microglia.

5. The authors show an interesting microglial phenotype that could explain a causal role of microglia in early stages of NPC disease. Interestingly, several markers identified from the proteomic analysis (P7) show overlap with downstream targets affected by HPβCD treatment (CD68, NPC2, CTSB, CTSD). Additionally, a previous study shows that HPβCD treatment in an earlier time period (presymptomatic stage) in NPC1-deficient mice is more effective in terms of reducing microglial activation and improving lifespan than later stage administration (*Hum Mol Genet*, 27(12), 2018, 2076–2089). Given the authors have potential protein markers associated with DAM phenotype in early stages of the disease model, the efficacy of HPβCD treatment or other pharmacological approach related to rescuing phenotype based on proteomic data would provide more value to this study.

We are thankful to the reviewer for this comment. In the revised manuscript we have examined the potential of methyl-β-cyclodextrin (MβCD) treatment in rescuing molecular and functional (endosomal/lysosomal trafficking) alterations of *Npc1*^{-/-} microglia. We observed a partial rescue of lipid droplet formation upon treatment of cultured *Npc1*^{-/-} microglia with MβCD (Fig. 7a). Next, we tested whether this functional rescue is also reflected by alterations in microglial signatures. Indeed, we detected increased levels of the key regulator of microglial homeostasis TGFB1 upon MβCD treatment and reduced levels of the DAM protein CTSD (Fig. 7b), strongly supporting that a switch towards more homeostatic microglial profiles underscores the rescue of the lipid droplet phenotype. Fully in line with our results, transcriptomic characterization of cyclodextrin treated NPC-deficient mice *in vivo* suggested improved homeostasis and reduced activation of *Npc1*^{-/-} microglia¹⁴.

Reviewer #2

1. Human gene nomenclature should be used when referring to the human gene. NPC1 and NPC2 not *Npc1* and *Npc2*. The latter being mouse nomenclature.

We thank the reviewer for pointing out the correct gene nomenclature. This has been changed accordingly in the revised manuscript.

2. Colocalization of filipin with lamp1 would be more typical than using CD68 to demonstrated late endosome/lysosomal localization.

We appreciate this comment. We initially tested several endosomal/lysosomal markers in microglia, including LAMP1 (direct comparison between LAMP1 and CD68 is shown in Figure for reviewer's assessment 2). As we were very limited with the number of microglial cells that can be isolated, we proceeded with using CD68 that is, in contrast to LAMP1, a bona fide endosomal/lysosomal marker in myeloid cells and its increased expression correlates with microglial activation and increased phagocytic capacity, phenotypes that were of interest for our study^{7,20}.

3. Previous work (Peake 2011 reference) has shown that Npc1 mutant microglia still display a cholesterol storage phenotype.

We have acknowledged this contribution (lines 129-131). To improve the novelty of our findings, we now include the analysis of the conditional NPC1 mouse model that lacks NPC1 only in microglia (*Npc1^{fllox/cre}*). We demonstrate a cell autonomous effect of NPC1 in regulating microglial endosomal/lysosomal homeostasis.

4. Line 178. Authors should address overlap of their findings with those previously described by Alam et al JBC 2014 289:8051-8066.

As suggested by the reviewer, we have discussed these findings along lines 421-424 of the revised manuscript.

5. Line 211. Authors should address prior work describing increased myelin uptake Gabande-Rodriguez EMBO J 2019 E99553.

Increased myelin uptake was discussed in the revised manuscript (lines 247-248).

6. 1/100,000 is a better estimate of live birth rate.

This has been modified accordingly (line 337).

7. Not clear how one obtains a "good correlation" when human samples are limited to an n of 2. Figure 7A only shows a change with age yet the text talks about pathological stage (what exactly are they referring to) and clinical symptoms (which ones). Describing NPC1 disease progression as typical is not useful. What are the potential effects of the concomitant medications on the proteomic study? Why was the validation only done with one of the two patient samples?

We agree with the reviewer that our initial sample size for the proteomic analysis was small (n=2). Accordingly, in the revised manuscript, we increased the number of analyzed patient samples to n=7. We also provide a better description of the clinical picture (severity grade) of NPC patients and their therapy (miglustat) that is included into the new Supplementary Table 5. We also agree with the reviewer that that medication may affect the study outcome, but as miglustat is approved in EU as a therapy for NPC, all of the patients included into this study received the drug and, unfortunately, we could not get any access to NPC patients that receive no medication. We discuss this limitation of our study along lines 418-420.

8. The idea that synaptic pruning is impaired needs further support.

As already outlined in our response to reviewer #1 (point 4), we agree that synaptic pruning defects in NPC need further mechanistic support. As we focused our revised manuscript in a way that synaptic pruning phenotype was not studied in more detail, we have toned down our statement that NPC1

regulates synaptic pruning. However, what our data clearly show is that *Npc1*^{-/-} microglia uptake neuronal material more readily, as they also do with myelin and A β substrates, further supporting their enhanced uptake capacity, as also seen in other neurodegenerative conditions^{17,18} and upon enhanced lysosomal activity¹⁹. Along these lines, microglial uptake of dendritic material has been detected prior to obvious neuronal loss¹⁵ and reduced spine density has also been reported in NPC-deficient mice *in vivo*¹⁶. This is discussed along lines 432-443 of our revised manuscript.

9. Line 299. It is not “striking” to find similarities between macrophage and microglia

We agree with the reviewer and have modified the phrase accordingly (line 363; “*strikingly*” has been replaced with “*in conclusion*”). We are also thankful to this reviewer that emphasized similarities between macrophages and microglia, supporting the idea that peripheral macrophages can be explored to study human NPC pathology in the brain.

10. Only one healthy control is used and that control is significantly older than either of the patients.

To address the reviewer’s criticism, we increased our control sample size to n=3. The 2 additional healthy volunteers are better matching the age of NPC patients (Supplementary Table 5).

11. Lack of perfusion could lead to contamination of the microglial preparation with peripheral mononuclear cells.

We agree with the reviewer that we cannot exclude minor contamination of microglial preparations by peripheral mononuclear cells. However, we estimate that this limitation would compromise all samples in a comparable manner. Furthermore, in the revised manuscript, we show complete list of all identified proteins (Supplementary Tables 1-4) which demonstrates high expression of the microglia specific genes in our proteomic datasets, suggesting that blood borne monocyte contamination of our samples is negligible. Additional quality controls for microglial isolation procedure and purity were also included in our recent manuscript that focused on the establishment of the method¹.

12. Is a student t-test appropriate for experiments with n=3? Would prefer authors to show SD rather than SEM. Would prefer authors show actual data points with the bar graphs. Provide information as to what statistical program was used.

It is suggested that t-test can also be used with small sample sizes, such as n=3²¹. In our data analysis, we were more interested in the precision of the means and in estimating differences between individual experimental groups. Thus, we preferred using standard error of the mean (SEM) instead of standard deviation (SD). In contrast to SD, SEM does not focus on the spread and variability of the data. In order to provide this additional information, individual data points are now added to all bar graphs. Statistical analysis is outlined in the Methods section of the manuscript (lines 761-765) and indicated in corresponding figure legends.

13. Abstract. The consequences of loss of NPC1 function on microglia may not be fully described, but it is not unknown.

This statement has been modified accordingly (lines 32-33).

14. The authors should discuss why the clearance rates of AB and myelin are different.

We have discussed this issue along lines 462-472 of the revised manuscript.

Work that should be addressed in more detail.

We have addressed this work in more detail as outlined below.

<https://www.ncbi.nlm.nih.gov/pmc/articles/PMC6331723/> (lines 89-95, 247-248, 375-376, 446-447 and 462-463)

<https://www.ncbi.nlm.nih.gov/pmc/articles/PMC3308731/> (lines 487-489)

<https://www.ncbi.nlm.nih.gov/pmc/articles/PMC5985727/> (lines 87-89, 93-95, 118-120, 150-152, 319-321, 373-375, 376-377, 385-389, 429-431, 435-437, 489-492)

<https://www.ncbi.nlm.nih.gov/pmc/articles/PMC5532464/> (lines 469-470, 492-495)

<https://www.ncbi.nlm.nih.gov/pmc/articles/PMC3961638/> (lines 421-422)

Reviewer #3

1. This is quite an interesting study, but highly descriptive with very little mechanistic insights. Moreover, since this is a general model of *Npc1*^{-/-} gene deletion, how the authors can be sure that protein changes in microglia are directly link to *Npc1* in microglia and not other cell types (Oligodendrocytes)?

It is generally well accepted that lysosomal function is impaired in NPC. Most of the conclusions supporting this idea were generated in cell lines or neuronal cultures. Our work focuses on the role of NPC1 in microglia and demonstrates distinct and substrate-specific endosomal/lysosomal trafficking defects upon NPC1 loss. We show that lipid substrates accumulate within multivesicular bodies (MVBs) and fail to reach lysosomes. This finding supports the view that primary defect in NPC is represented by aberrant intracellular trafficking that is preventing the lipid substrate to reach lysosomes. In contrast, degradation of protein substrates, such as A β , was rather enhanced in *Npc1*^{-/-} microglia. Thus, our data do not support the hypothesis that degradation function of lysosomes in microglia is impaired upon loss of NPC1 and this is a novel finding. To provide additional mechanistic support for this hypothesis, we included in the revised manuscript an additional protein substrate degradation assay (EGFR degradation) that fully supports preserved function of lysosomal degradation in *Npc1*^{-/-} microglia. In accordance with this, we observed normal maturation of CTSB or CTSD, and this process relies on functional lysosomal activity. We now also provide additional mechanistic evidence that aberrant lipid droplet formation is mediated by increased levels of cholesterol in *Npc1*^{-/-} microglia as this phenotype could be partially rescued by lowering cholesterol using methyl- β -cyclodextrin (M β CD) (Fig. 7a). Furthermore, this functional rescue is also reflected by alterations in microglial signatures. We detected increased levels of the key regulator of microglial homeostasis TGFB1 and reduced levels of the DAM protein CTSD (Fig. 7b) upon M β CD treatment, strongly supporting that a switch towards more homeostatic microglial profiles underscores the rescue of the lipid droplet phenotype.

We fully agree with the reviewer's criticism that microglial phenotypes described using the full *Npc1*^{-/-} mouse model may be triggered by combination of cell autonomous defects in microglia and lack of NPC1 in other pathologically relevant cells, including neurons, astrocytes and oligodendrocytes. To address this question and increase the novelty of our work, we generated a conditional mouse model that lacks NPC1 only in myeloid cells (*Npc1*^{fl α /cre}). We showed for the first time that loss of NPC1 in microglia only is sufficient to trigger molecular changes and morphological transformation of ramified into fully activated amoeboid microglia. This data are included into the new Figure 3. Moreover, we find a substantial overlap of proteomic signatures in *Npc1*^{fl α /cre} and *Npc1*^{-/-} mice (Supplementary Fig. 3c), supporting that loss of NPC1 from microglia is sufficient to trigger increased DAM and reduced homeostatic signatures of microglia. We also provide evidence for myelin (Fig. 3d, Fig. 4d and e) and synaptophysin (Fig. 3d) accumulation within microglia of *Npc1*^{fl α /cre} mice, strongly supporting cell autonomous role of NPC1 in microglia.

2. The data are also a little bit all over the place from Abeta and myelin phagocytosis to synaptic pruning. The link between these two events are not clear, since the clearance of toxic proteins would normally be beneficial for neurons and pruning!

We agree with the reviewer that clearance of toxic proteins is a beneficial process. One can speculate that A β accumulation in NPC could be even more pronounced if microglial clearance would not be that efficient. However, also detrimental effects for neurons may be triggered by exaggerated phagocytic activity of microglia as shown in other neurodegenerative diseases¹⁷⁻¹⁹.

3. In addition, the proteomic data are largely descriptive and do not provide direct mechanistic evident that these changes are related to the phagocytosis and pruning by microglia and human macrophages. Obviously, macrophages would not be involved in synaptic pruning!

As largely outlined above, we have included additional mechanistic and pathway analysis (new Fig. 7, Supplementary Fig. 2 and 3) and characterization of the new mouse model *Npc1*^{flox/cre} (new Fig. 3, Fig. 4d and e) in the revised version of the manuscript. Of note, we fully agree with the reviewer that macrophages and microglia see different physiological substrates, but, as also acknowledged by the reviewer #2 (point 9), these cell types also share similarities and thus peripheral macrophages can serve as valuable tool to reveal disease mechanisms in the brain²².

Literature:

- 1 Monasor, L. S. *et al.* Fibrillar A β triggers microglial proteome alterations and dysfunction in Alzheimer mouse models. *eLife in press*, doi:10.1101/861146 (2020).
- 2 Tusher, V. G., Tibshirani, R. & Chu, G. Significance analysis of microarrays applied to the ionizing radiation response. *Proc Natl Acad Sci U S A* **98**, 5116-5121, doi:10.1073/pnas.091062498 (2001).
- 3 Pascovici, D., Handler, D. C., Wu, J. X. & Haynes, P. A. Multiple testing corrections in quantitative proteomics: A useful but blunt tool. *Proteomics* **16**, 2448-2453, doi:10.1002/pmic.201600044 (2016).
- 4 Cermak, S. *et al.* Loss of Cathepsin B and L Leads to Lysosomal Dysfunction, NPC-Like Cholesterol Sequestration and Accumulation of the Key Alzheimer's Proteins. *PLoS One* **11**, e0167428, doi:10.1371/journal.pone.0167428 (2016).
- 5 Roszell, B. R. *et al.* Pulmonary abnormalities in animal models due to Niemann-Pick type C1 (NPC1) or C2 (NPC2) disease. *PLoS One* **8**, e67084, doi:10.1371/journal.pone.0067084 (2013).
- 6 Buschow, S. I. *et al.* Unraveling the human dendritic cell phagosome proteome by organellar enrichment ranking. *J Proteomics* **75**, 1547-1562, doi:10.1016/j.jprot.2011.11.024 (2012).
- 7 Daria, A. *et al.* Young microglia restore amyloid plaque clearance of aged microglia. *EMBO J* **36**, 583-603, doi:10.15252/embj.201694591 (2017).
- 8 Song, L. *et al.* KIBRA controls exosome secretion via inhibiting the proteasomal degradation of Rab27a. *Nat Commun* **10**, 1639, doi:10.1038/s41467-019-09720-x (2019).
- 9 Gotzl, J. K. *et al.* Common pathobiochemical hallmarks of progranulin-associated frontotemporal lobar degeneration and neuronal ceroid lipofuscinosis. *Acta Neuropathol* **127**, 845-860, doi:10.1007/s00401-014-1262-6 (2014).
- 10 Wu, Y. *et al.* Blastomere biopsy influences epigenetic reprogramming during early embryo development, which impacts neural development and function in resulting mice. *Cell Mol Life Sci* **71**, 1761-1774, doi:10.1007/s00018-013-1466-2 (2014).
- 11 Beaufort, N. *et al.* Cerebral small vessel disease-related protease HtrA1 processes latent TGF-beta binding protein 1 and facilitates TGF-beta signaling. *Proc Natl Acad Sci U S A* **111**, 16496-16501, doi:10.1073/pnas.1418087111 (2014).

- 12 Xiang, X. *et al.* TREM2 deficiency reduces the efficacy of immunotherapeutic amyloid clearance. *EMBO Mol Med* **8**, 992-1004, doi:10.15252/emmm.201606370 (2016).
- 13 Gotzl, J. K. *et al.* Early lysosomal maturation deficits in microglia triggers enhanced lysosomal activity in other brain cells of progranulin knockout mice. *Mol Neurodegener* **13**, 48, doi:10.1186/s13024-018-0281-5 (2018).
- 14 Cougnoux, A. *et al.* Microglia activation in Niemann-Pick disease, type C1 is amendable to therapeutic intervention. *Hum Mol Genet* **27**, 2076-2089, doi:10.1093/hmg/ddy112 (2018).
- 15 Kavetsky, L. *et al.* Increased interactions and engulfment of dendrites by microglia precede Purkinje cell degeneration in a mouse model of Niemann Pick Type-C. *Sci Rep* **9**, 14722, doi:10.1038/s41598-019-51246-1 (2019).
- 16 Tiscione, S. A. *et al.* Disease-associated mutations in Niemann-Pick type C1 alter ER calcium signaling and neuronal plasticity. *J Cell Biol* **218**, 4141-4156, doi:10.1083/jcb.201903018 (2019).
- 17 Filipello, F. *et al.* The Microglial Innate Immune Receptor TREM2 Is Required for Synapse Elimination and Normal Brain Connectivity. *Immunity* **48**, 979-991 e978, doi:10.1016/j.immuni.2018.04.016 (2018).
- 18 Hong, S. *et al.* Complement and microglia mediate early synapse loss in Alzheimer mouse models. *Science*, doi:10.1126/science.aad8373 (2016).
- 19 Lui, H. *et al.* Progranulin Deficiency Promotes Circuit-Specific Synaptic Pruning by Microglia via Complement Activation. *Cell* **165**, 921-935, doi:10.1016/j.cell.2016.04.001 (2016).
- 20 Chistiakov, D. A., Killingsworth, M. C., Myasoedova, V. A., Orekhov, A. N. & Bobryshev, Y. V. CD68/macrosialin: not just a histochemical marker. *Lab Invest* **97**, 4-13, doi:10.1038/labinvest.2016.116 (2017).
- 21 de Winter, J. C. F. Using the Student's t-test with extremely small sample sizes. *Practical Assessment, Research, and Evaluation* **Vol. 18**, doi:https://doi.org/10.7275/e4r6-dj05 (2013).
- 22 Borger, D. K., Sidransky, E. & Aflaki, E. New macrophage models of Gaucher disease offer new tools for drug development. *Macrophage (Houst)* **2**, e712 (2015).

REVIEWER COMMENTS

Reviewer #1 (Remarks to the Author):

The authors have adequately addressed several points brought up in the original review including more attention to interpretation of the bioinformatic results and related functional outcomes measured experimentally; however, certain issues still remain and are described below:

1) In regard to the proteomic data/statistical analysis, the selection of the s_0 value/FDR parameters is still not clear. The original paper describing the approach (and cited within Perseus) was provided; however, the relevance of the filtering was not provided. My concern is shown in Fig. 1B (and 3D) where the \log_2 ratio shows very low ($< < 0.5$) cutoff at $-\log(p\text{-value}) > 1.3$ for the FDR curve ($p < 0.05$ and $s_0 = 0.1$), indicating that most proteins will achieve this FDR cutoff essentially at the $p < 0.05$ mark (which would appear to have minimal FDR control). Looking specifically at the supplemental table for 8 week comparison, for example, < -0.18 and > 0.17 \log_2 change was the cutoff for down- and up-regulated, respectively, at $p < 0.05$ to achieve this FDR. More clarification is needed to ensure adequate FDR and ultimately reporting of reproducible markers associated with loss of NPC1 in microglia.

2) LFQ intensities were requested for the proteins identified and were not provided. The intensities can be useful to determine relative contribution of individual proteins (for example, from contaminating cell types/debris) to overall protein intensity for each group and rep. For example, hemoglobin and GFAP were identified and intensity level, if reported, could indicate a more quantitative assessment of potential blood cell and astrocyte contamination, respectively.

3) The authors toned down on synaptic pruning in NPC in relation to this study and provide literature support in this regard. Minor point: related to the proteomic analysis (and somewhat to point 2), a single marker was used, SYP, to indicate microglial uptake. Given there are other cell type-"specific" proteins rather than from microglia (for example, more synaptic proteins other than SYP), could the authors perform further potential substrate classification of the proteins within the proteome datasets - this could strengthen argument and allay concern regarding contamination (page 9).

Reviewer #3 (Remarks to the Author):

The revised version is appropriate and all raised comments were addressed.

Point-by-point response to reviewers' comments on the manuscript "Loss of NPC1 enhances phagocytic uptake and impairs lipid trafficking in microglia" (NCOMMS-19-27509A-Z)

We would particularly like to thank to both reviewers for acknowledging our efforts to improve the manuscript. Please, find below our point-by-point response to the remaining concerns raised by the reviewer 1. Changes to the manuscript are marked in red.

Reviewer 1:

1) In regard to the proteomic data/statistical analysis, the selection of the s0 value/FDR parameters is still not clear. The original paper describing the approach (and cited within Perseus) was provided; however, the relevance of the filtering was not provided. My concern is shown in Fig. 1B (and 3D) where the log2ratio shows very low (<<0.5) cutoff at $-\log(p\text{-value}) > 1.3$ for the FDR curve ($p < 0.05$ and $s0 = 0.1$), indicating that most proteins will achieve this FDR cutoff essentially at the $p < 0.05$ mark (which would appear to have minimal FDR control). Looking specifically at the supplemental table for 8 week comparison, for example, < -0.18 and > 0.17 log2 change was the cutoff for down- and up-regulated, respectively, at $p < 0.05$ to achieve this FDR. More clarification is needed to ensure adequate FDR and ultimately reporting of reproducible markers associated with loss of NPC1 in microglia.

We acknowledge the criticism that the selected fixed s0 value of 0.1 for the FDR control might not be optimal. To ensure the adequate FDR cutoff, we now experimentally determined the s0 values separately for each experiment. Briefly, according to published equations one and two on page 5117¹, the "relative difference" d_i of each protein was calculated, which includes s0 as a variable. Next, s0 was optimized separately for each experimental dataset, based on an interpolation to reach the lowest coefficient of variation for the absolute d_i values. This resulted in a s0 value of 0.37 for *Npc1*^{-/-} vs WT microglia at 8 weeks (Figure 1b), a s0 value of 0.19 for *Npc1*^{-/-} vs WT microglia at P7 (Figure 2c), a s0 value of 0.18 for *Cre+* vs *Cre-* microglia (Figure 3d) and a s0 value of 0.37 for human NPC1 mutation carriers vs the control group (Figure 8a). Noteworthy, experimentally determined new s0 values do not affect our reported Ingenuity Pathway Analysis and data interpretation as we only considered proteins with a log2 LFQ ratio $> +/- 0.5$ and a p-value < 0.05 for the analysis. To highlight the applied thresholds, we labeled proteins with a p-value less than 0.05, a log2 ratio larger than $+/- 0.5$ and significance after the FDR correction as black circles in all volcano plots. We are thankful to the reviewer for motivating us to improve the data presentation and thus report more robust and reproducible markers of NPC1 loss of function in myeloid cells.

2) LFQ intensities were requested for the proteins identified and were not provided. The intensities can be useful to determine relative contribution of individual proteins (for example, from contaminating cell types/debris) to overall protein intensity for each group and rep. For example, hemoglobin and GFAP were identified and intensity level, if reported, could indicate a more quantitative assessment of potential blood cell and astrocyte contamination, respectively.

We agree with the reviewer that it is of relevance to provide the LFQ intensities. We have initially only included them as raw data accessible at the PRIDE archive. In the revised manuscript, we added LFQ intensities also into Supplementary Tables 1-4.

3) The authors toned down on synaptic pruning in NPC in relation to this study and provide literature support in this regard. Minor point: related to the proteomic analysis (and somewhat to point 2), a single marker was used, SYP, to indicate microglial uptake. Given there are other cell type-"specific" proteins rather than from microglia (for example, more synaptic proteins other than SYP), could the

authors perform further potential substrate classification of the proteins within the proteome datasets - this could strengthen argument and allay concern regarding contamination (page 9).

We fully agree with the reviewer that –in addition to microglial– other cell type specific proteins are also detected in our proteomic analysis. We believe that, in particular myelin specific or neuronal/synaptic proteins may be detected due to the enhanced phagocytic uptake by microglia as shown for MBP, SYP or Hoechst-positive nuclei. However, we cannot exclude possible contamination of the microglial fraction with proteins from other brain cells. We have now classified protein signatures according to reported cellular origin ² (new Supplementary Tables 1-3), and reveal an enrichment of microglial proteins in our datasets. We would also like to note that additional characterization of the microglia-enriched fraction by FACS analysis (using CD11b marker) revealed a 97% of CD11b positive cells in microglia-enriched fraction ³, supporting relatively high purity of our cell isolation procedure.

Literature:

- 1 Tusher, V. G., Tibshirani, R. & Chu, G. Significance analysis of microarrays applied to the ionizing radiation response. *Proc Natl Acad Sci U S A* **98**, 5116-5121, doi:10.1073/pnas.091062498 (2001).
- 2 Sharma, K. *et al.* Cell type- and brain region-resolved mouse brain proteome. *Nat Neurosci* **18**, 1819-1831, doi:10.1038/nn.4160 (2015).
- 3 Sebastian Monasor, L. *et al.* Fibrillar Abeta triggers microglial proteome alterations and dysfunction in Alzheimer mouse models. *Elife* **9**, doi:10.7554/eLife.54083 (2020).

REVIEWER COMMENTS

Reviewer #1 (Remarks to the Author):

The authors have improved the analysis and presentation of the proteomic data. All concerns have been addressed.